# Sanity Checking Causal Representation Learning
# on a Simple Real-World System

**Juan L. Gamella** [* 1]  **Simon Bing** [* 2 3]  **Jakob Runge** [4 3]

## Abstract

We evaluate methods for causal representation learning (CRL) on a simple, real-world system where these methods are expected to work. The system consists of a controlled optical experiment specifically built for this purpose, which satisfies the core assumptions of CRL and where the underlying causal factors—the inputs to the experiment—are known, providing a ground truth. We select methods representative of different approaches to CRL and find that they all fail to recover the underlying causal factors. To understand the failure modes of the evaluated algorithms, we perform an ablation on the data by substituting the real data-generating process with a simpler synthetic equivalent. The results reveal a reproducibility problem, as most methods already fail on this synthetic ablation despite its simple data-generating process. Additionally, we observe that common assumptions on the mixing function are crucial for the performance of some of the methods but do not hold in the real data. Our efforts highlight the contrast between the theoretical promise of the state of the art and the challenges in its application. We hope the benchmark serves as a simple, real-world sanity check to further develop and validate methodology, bridging the gap towards CRL methods that work in practice.

## 1. Introduction

Uncovering the underlying factors that determine the behavior of a system is a problem with a long history, both in

---
[*]Equal contribution  [1]Seminar for Statistics, ETH Zurich [2]Technische Universität Berlin [3]Department of Computer Science, University of Potsdam [4]ScaDS.AI Dresden/Leipzig, TU Dresden. Correspondence to: Juan L. Gamella <juangamella@gmail.com>, Simon Bing <bing@campus.tu-berlin.de>.

*Proceedings of the $42^{nd}$ International Conference on Machine Learning*, Vancouver, Canada. PMLR 267, 2025. Copyright 2025 by the author(s).

theoretical treatise (Comon, 1994; Hyvärinen & Pajunen, 1999; Bengio et al., 2013; LeCun et al., 2015; Lake et al., 2017), as well as practical applications (Johansson et al., 2022; Tibau et al., 2022; Lopez et al., 2023). Causal representation learning is among the newest approaches in this line of work, and its enticing promise lies in recovering the *causal* latent ground-truth model from high-level observations of a system. While this problem is provably hard (Locatello et al., 2019), there has been a considerable effort to advance the understanding of the theoretical constraints within which causal representation learning may be successfully applied (Hyvärinen & Morioka, 2016; Khemakhem et al., 2020; Gresele et al., 2021; Brehmer et al., 2022; Ahuja et al., 2023; Zhang et al., 2023; Lippe et al., 2022; 2023a;b; Lachapelle et al., 2023; Varıcı et al., 2023; Saengkyongam et al., 2024; von Kügelgen et al., 2024; Yao et al., 2024b). Applying established causal inference methods to unstructured, high-dimensional data, addressing the inherent limitations of machine learning methods in out-of-distribution scenarios, or learning mechanistic models of the world are among the multitude of promises that drive research in the field of causal representation learning (Schölkopf et al., 2021).

A challenge that obfuscates the progress towards these goals is the lack of meaningful real-world benchmarks, to evaluate methods and identify theoretical approaches that hold potential for application. By and large, new methods are evaluated on synthetic datasets generated according to their own underlying assumptions (Brehmer et al., 2022; Ahuja et al., 2023; Lippe et al., 2022; 2023a;b; Squires et al., 2023; Lachapelle et al., 2023; 2024; Liang et al., 2023; Buchholz et al., 2023; Varıcı et al., 2023; Bing et al., 2024; von Kügelgen et al., 2021; von Kügelgen et al., 2024; Yao et al., 2024c; Xu et al., 2024). This practice provides further validation for the theoretical foundations of these methods but yields limited insight into their applicability to real-world problems. More sophisticated synthetic benchmarks have been proposed (Lippe et al., 2022; 2023a; Ahmed et al., 2021; von Kügelgen et al., 2021; Liu et al., 2023), e.g., using visualizations of computer games or renderings of three-dimensional scenes. Because these do not cater to the assumptions of any particular method, they are extremely valuable for the standardized evaluation of CRL methods.

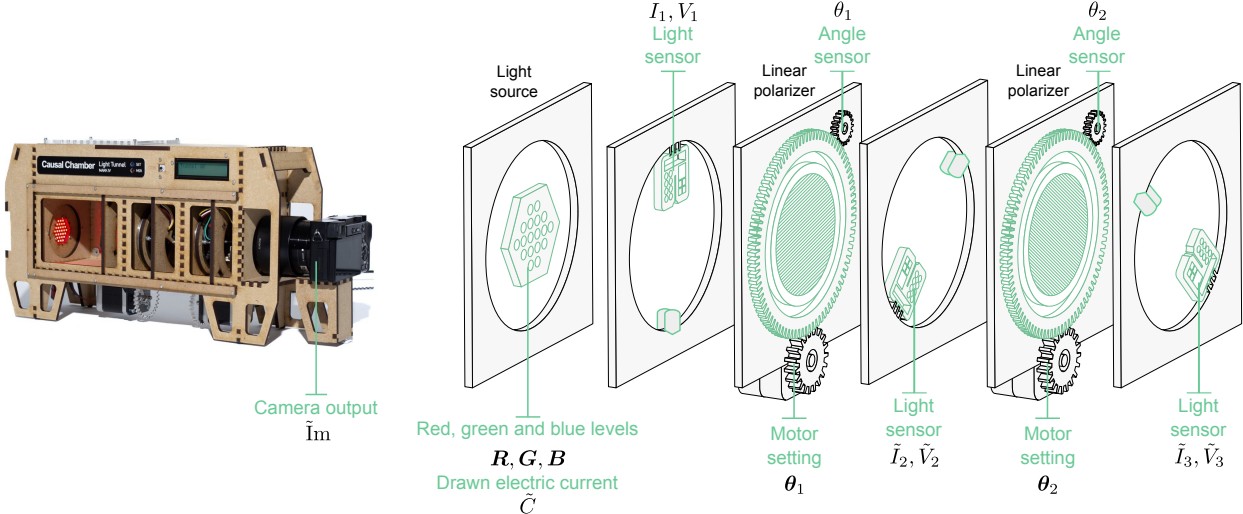

Figure 1: The light tunnel (left) and a simplified schematic (right) showing its main components and variables. The tunnel consists of a controllable light source, linear polarizers mounted on rotating frames, a camera, and sensors to measure light intensity at different wavelengths and positions. The inputs to the system $(R, G, B, \theta_1, \theta_2)$ are displayed in bold math print in this figure. The system's outputs—image data and numeric sensor measurements—are denoted by a tilde.

However, their synthetic nature means they are limited as testbeds to discuss which CRL approaches—and their underlying theory—are applicable in the real world. While real-world datasets are often used to illustrate potential applications of new methods (Zhang et al., 2023; Chen et al., 2024; Yao et al., 2024a), the absence of a ground truth also precludes any in-depth analysis into this matter.

As an orthogonal contribution to existing synthetic benchmarks, we propose a real-world sanity check to test the applicability of CRL methods and their underlying theory. The test is based on a carefully designed, real, physical system whose data-generating process matches the core assumptions of causal representation learning (Section 2). Furthermore, the relationship between latent factors and observations is far simpler than in the synthetic benchmarks mentioned in the previous paragraph. Thus, we base our sanity check on the premise that a generic CRL method—designed to work in various settings—should also work on this simple but real system.

The physical system consists of a well-understood optical experiment involving the polarization of light. The system was first introduced by Gamella et al. (2025) and is described in detail in Section 2. The control inputs to the experiment constitute the underlying causal factors, and its outputs—image and sensor data—correspond to the entangled observations. Because the control inputs are known, we have a ground truth to directly evaluate the performance of the tested methods in recovering the underlying causal factors.

As a first application of our sanity check, we evaluate a representative method from three different families of approaches: CRL methods requiring interventional (or counterfactual) data (Brehmer et al., 2022; Ahuja et al., 2023; Zhang et al., 2023; Squires et al., 2023; Buchholz et al., 2023; Varıcı et al., 2023; Liang et al., 2023; Bing et al., 2024; von Kügelgen et al., 2024), CRL considering multiple views (Gresele* et al., 2019; Locatello et al., 2020; von Kügelgen et al., 2021; Daunhawer et al., 2023; Ahuja et al., 2024; Xu et al., 2024; Yao et al., 2024c), and CRL based on time-series data (Yao et al., 2021; 2022; Lippe et al., 2022; 2023a;b; Lachapelle et al., 2024). The results and background needed to understand them are given for each method in Section 3.

To our knowledge, this sanity check is the first of its kind, and we hope that it can serve as a standardized benchmark to evaluate new methods and test the assumptions that underlie theoretical work. However, we find it important to clearly delimit the conclusions that can be drawn from this benchmark. First, our current setup is geared towards evaluating the assumptions concerning the mixing function, as they underpin the identifiability results of many methods, and they are the most straightforward to test. This means that the applicability of other assumptions, for example, regarding modeling the latent causal structure using a structural causal model (SCM, Peters et al., 2017, Chapter 3), cannot be readily assessed using the datasets we currently provide. We further discuss this point in Section 2. Second, as with any test, only a negative result is truly informative: a method

that fails on this simple and tightly controlled testbed can be expected to fail in other, more complex, real-world scenarios. However, the converse does not hold. While our setup is representative of other systems that produce data with digital sensors, a method that succeeds on our benchmark is not guaranteed to do so on all other real-world systems that are the target of application.

The physical system used for our benchmark is one of the two Causal Chambers first introduced by Gamella et al. (2025). Our contribution is to leverage this system to develop a meaningful sanity check for CRL algorithms. We make the novel datasets and their data-collection procedures publicly available in the `lt_crl_benchmark_v1` dataset at `github.com/juangamella/causal-chamber`. The code to reproduce the results of this paper can be found at `github.com/simonbing/CRLSanityCheck`. The deterministic simulators developed for the synthetic ablation (Section 2) are documented in Appendix C, and we provide a Python implementation in the `causalchamber` package.

## 2. Experimental Setup

Our physical system is a light tunnel like the one introduced by Gamella et al. (2025). We provide a schematic of its main components and relevant variables in Figure 1. The tunnel is an elongated chamber with a controllable light source at one end, two linear polarizers mounted on rotating frames, a camera, and sensors to measure different physical quantities. As control inputs, the system takes the brightness level of the red, green, and blue LEDs of the light source $(R, G, B)$ and the polarizer positions $(\theta_1, \theta_2)$. Its outputs are the images captured by the camera (Figure 2) and readings of the infrared $(\tilde{I}_1, \tilde{I}_2, \tilde{I}_3)$ and visible $(\tilde{V}_1, \tilde{V}_2, \tilde{V}_3)$ light intensity at different positions, the electrical current drawn by the light source $(\tilde{C})$, and additional noisy measurements of the polarizer angles $(\tilde{\theta}_1, \tilde{\theta}_2)$. A detailed description of all the variables can be found in Gamella et al. (2025, Appendix II).

As a data-generating process, the light tunnel satisfies the core assumptions of causal representation learning. In particular, it transforms some underlying causal factors—the control inputs—into observations consisting of images and numeric sensor measurements. Because we control the inputs to the system, we can sample them from any distribution or causal structure, as we do in the experiments of Section 3. However, this means that the underlying causal structure is synthetic, and the real-world applicability of the corresponding assumptions cannot be evaluated. Therefore, using the current setup we can only test the assumptions corresponding to the mixing function, but not those placed on the latent causal model generating the data.

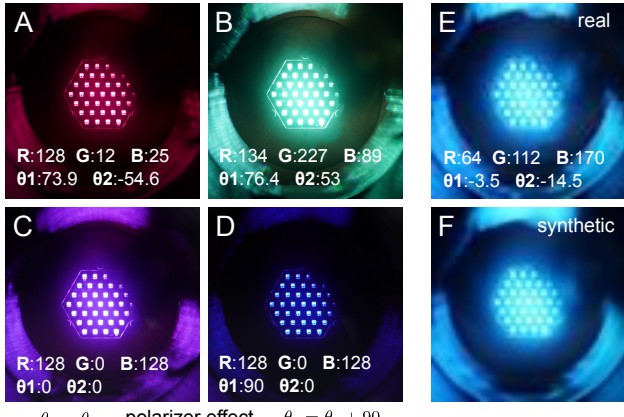

Figure 2: **(A–D):** Real images collected from the light tunnel, with the corresponding control inputs overlaid in white. The light source color $(R, G, B)$ is the same for images C and D, but the first polarizer is shifted by 90 degrees in image D, showing the effect of the linear polarizers angles $(\theta_1, \theta_2)$. **(E, F):** Comparison between (E) a real image (at low resolution) and (F) the synthetic counterpart produced by a simple multi-layer perceptron given the same inputs.

**A Real-World Mixing Function.** The light tunnel provides an ideal testbed to evaluate assumptions concerning the mixing function that transforms the ground-truth causal factors into observations. In the tunnel, the process that produces the images is far simpler than those in common synthetic benchmarks, which arise, for example, from visualizations of computer games (Lippe et al., 2022; 2023a) or renderings of a physical simulator (Ahmed et al., 2021; von Kügelgen et al., 2021). Evidence of this simplicity is that the process can be simulated to a high degree of visual realism with a simple multi-layer perceptron (MLP) (Figure 2, E/F). Furthermore, the relationship between the light source setting $(R, G, B)$ and the pixel intensities is approximately linear up to the overexposure of the camera sensor, as happens in the central region of the images. The polarizer angles $(\theta_1, \theta_2)$ scale the light intensity reaching the camera by a factor of $\cos^2(\theta_1 - \theta_2)$, as dictated by Malus' law (Collett, 2005). This dimming affects each color wavelength differently, resulting in a distortion of the color noticeable at relative angles close to $90°$ (see D/E in Figure 2). These effects are thoroughly documented in Gamella et al. (2025, Appendix III). The images also contain some other subtle artifacts, introduced by the chromatic aberration of the lens, the reflections of the polarizer frames, and the processor onboard the camera. Different inputs always result in different images.

A crucial difference with commonly used synthetic benchmarks is the noise present in the images and sensor measurements, which naturally arises from several processes. The

first is the fluctuation of the light source, which modulates its brightness by switching on and off at a frequency of 2KHz, interacting with the photodiodes on the camera and the sensors in a complex way. The second is the measurement noise introduced by the sensors themselves, which is to be expected from any real system. For the camera and the third light sensor, which are placed behind the polarizers, the relative strength of these two noise sources is modulated by the polarizer positions $\theta_1$ and $\theta_2$. As a result, the "mixing function" of the light tunnel is not a deterministic process. Furthermore, this stochasticity is not captured by an additive noise model.

**Synthetic Ablation.** To explore the effect that these subtle artifacts have on the performance of causal representation learning methods, we perform additional experiments where we substitute the real data-generating process of the light tunnel with a deterministic simulator that produces synthetic images and sensor measurements given the same inputs $(R, G, B, \theta_1, \theta_2)$. The simulator produces synthetic measurements via a mechanistic model and synthetic images (Figure 2F) via a simple MLP trained on a separate dataset. Further details about the simulator and its construction can be found in Appendix C.

## 3. Results

We apply our sanity check to methods representative of three different families of approaches: contrastive CRL (Section 3.1), multiview CRL (Section 3.2), and CRL from temporal intervened sequences (Section 3.3). As a result, the methods differ greatly in their setup and assumptions on the latent causal structure, as well as in their goals and the metrics to evaluate them. This makes a direct comparison between them difficult, and we instead perform our analysis method-by-method, introducing the necessary background together with the results.

In our efforts, we encounter a first challenge in the application of causal representation learning methods to real-world problems: a recurrent lack of public, well-documented, and tested code. As a result, our choice of methods is biased towards those with public code, or for which the authors provided code upon request.

We find that pre-processing steps and implementation decisions—such as network architectures and training procedures—have a drastic effect on the performance of some of the methods. This also highlights a potential difficulty in re-implementing CRL methods to reproduce their results. To minimize potential points of failure, we used the original implementations as much as possible. We thank the authors of the selected methods for their assistance, which was instrumental in getting them to run for this benchmark. To further ensure that a bug in our pipeline does not distort

our results, we ran preliminary tests on synthetic datasets replicating the settings used in the respective original works (Buchholz et al., 2023; Yao et al., 2024c; Lippe et al., 2022). Our experimental pipeline for each method is described in detail in Appendix A.

### 3.1. Contrastive CRL

The most prevalent class of CRL methods are those that assume access to data from different environments, stemming from interventions (or counterfactual observations) on some of the underlying causal factors (Brehmer et al., 2022; Ahuja et al., 2023; Zhang et al., 2023; Squires et al., 2023; Buchholz et al., 2023; Varıcı et al., 2023; Liang et al., 2023; Bing et al., 2024; von Kügelgen et al., 2024). We choose the method of Buchholz et al. (2023) as a representative of this family of models. In the remainder of the text, we will refer to it as Contrastive CRL (CCRL).

**Background.** As an underlying causal model, CCRL assumes a linear structural causal model (SCM) with additive Gaussian noise (Buchholz et al., 2023, Assumption 2). The underlying causal factors are transformed into observations through a nonlinear and deterministic mixing function (Buchholz et al., 2023, Assumption 1); while in theory their results can be extended to stochastic mixing functions with additive noise, both their experiments and implementation assume a deterministic mixing function. Provided with data from environments that correspond to a single-node intervention on each causal variable, the goal of the method is to recover the ground-truth causal factors and the graph encoding the causal structure between them (Buchholz et al., 2023, Theorem 1).

**Data Generation.** In line with the assumptions made by the method, we sample the values for the ground-truth factors—the tunnel inputs $R, G, B, \theta_1$ and $\theta_2$—from a linear SCM with additive Gaussian noise; the corresponding ground-truth graph is shown in Figure 3. To generate the interventional data, we perform interventions that shift the mean of their target, closely following the assumptions described in Buchholz et al. (2023, Assumption 3). We collect 10K observations per environment, constituting a total of 60K images provided as input to the method. A detailed exposition of the data-generating process is available in Appendix A.1.1.

**Results.** In Figure 3, we provide a summary of the results of applying CCRL to the real images and to the synthetic images (e.g., Figure 2F) produced by the deterministic simulator of the light tunnel described in Section 2 and Appendix C. We report the same metrics used in the original experiments of the method: the commonly used mean correlation coefficient (MCC, Hyvärinen & Morioka, 2016;

| | MCC ↑ | SHD ↓ |
|---|---|---|
| Real | $0.285 \pm 0.054$ | $7.600 \pm 0.894$ |
| Synth. ablation | $0.891 \pm 0.005$ | $2.000 \pm 0.000$ |

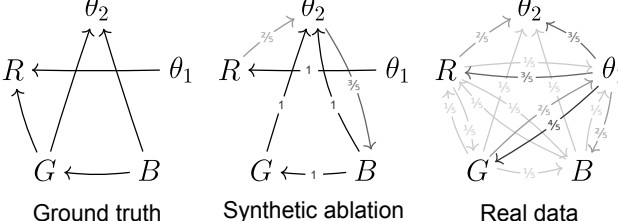

Figure 3: Experimental results for the Contrastive CRL method applied to the real data and the data from the synthetic ablation described in Section 2. **Left:** MCC score of predicting the ground-truth factors, and SHD score between the estimated latent graph and the ground-truth graph (shown on the left). We provide the average and standard deviation ($\pm$) of the scores over five random initializations of the method. **Right:** Ground-truth graph and a summary of the estimated latent graphs produced by five random initializations of the method. The edges are shaded and labeled according to the frequency they appear in the estimates obtained in the five runs of the method, with darker edges appearing more often. The method performs well on the data from the synthetic ablation, where a deterministic simulator (Appendix C) substitutes the data-generating process of the light tunnel. The method fails to produce meaningful results based on the real data.

Khemakhem et al., 2020), which is a measure of how well the method recovers the latent factors, and the structural Hamming distance (SHD, Tsamardinos et al., 2006) that measures the recovery of the latent causal graph.

The method obtains fairly good metrics on the data from the deterministic simulator, indicating that this is a setting where most necessary assumptions for CCRL are met, allowing the method to recover the latent variables and graph reasonably well. However, the method exhibits a stark drop in performance when applied to the real data, failing to recover the latent variables and returning vastly different latent graphs in each of its five random initializations. Since the main difference between these two settings is the determinism of the mixing function (see Section 2), we conclude that CCRL is highly sensitive to noise in the mixing process. A possible explanation is that CCRL relies on detecting interventions in the latent space, a sensitive statistical problem for which it may lack power in the case of a noisy mixing process.

### 3.2. Multiview CRL

The multiview approach to CRL (Gresele* et al., 2019; Locatello et al., 2020; von Kügelgen et al., 2021; Daunhawer et al., 2023; Ahuja et al., 2024; Xu et al., 2024; Yao et al., 2024c) relies on having different sets of observations, or views, that arise from mixtures of different subsets of the underlying causal factors. We choose the method introduced by Yao et al. (2024c) as a representative of this family of approaches.

**Background.** As opposed to CCRL, the method from Yao et al. (2024c) places more flexible assumptions on the distribution of the underlying causal factors, where any smooth,

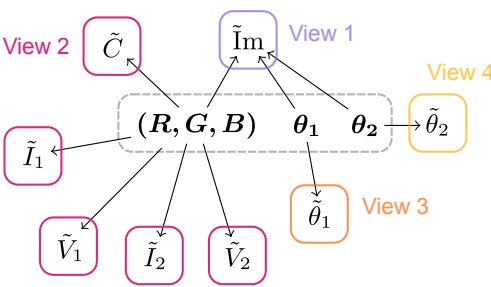

Figure 4: Ground-truth graph relating the underlying causal factors, shown in bold print, to the different views employed in the Multiview CRL experiment. The views are disjoint sets of the output variables produced by the tunnel. The separate factors $(R, G, B)$ are shown as a tuple to avoid drawing additional edges. The graph is a subgraph of the complete causal ground-truth graph for the light tunnel, described in Gamella et al. (2025, Figure 3b).

continuous distribution with a positive density almost everywhere is allowed (Yao et al., 2024c, Assumption 2.1). The underlying factors are transformed into observations via multiple diffeomorphic mixing functions that share (subsets of) these factors as their inputs, resulting in multiple views on the underlying causal model. For our testbed, we group the images and sensor measurements produced by the light tunnel into different views, as shown in Figure 4. Given a set of views, the goal of the method is to recover the underlying factors in the intersection of their inputs. These shared factors are referred to as the "content block" of a set of views, whereas those not in the intersection of inputs are called the "style block". Regarding identifiability, the theory guarantees partial (or block-) identifiability (Lachapelle

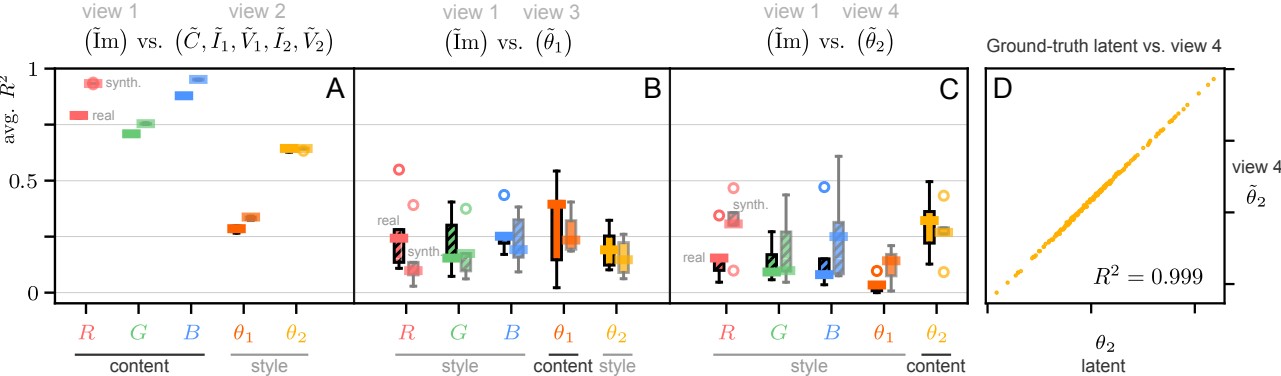

Figure 5: Experimental results for the multiview CRL method over five random initializations. **(A, B, C):** Average $R^2$-score of predicting the ground-truth factors from the learned representation across different pairs of views. Success is indicated by simultaneously attaining a score near one for the factors in the content block and a significantly lower score for those in the style block. While content variables are consistently predicted better than style variables, the model fails to disentangle $\theta_1$ and $\theta_2$, evident in panel B & C. **(D):** Scatter plot of the ground-truth factor $\theta_2$ vs. the corresponding sensor measurement in view 4 ($\tilde{\theta}_2$). Even though this view and the ground-truth factor are almost perfectly correlated, the model fails to recover the underlying factor $\theta_2$.

& Lacoste-Julien, 2022) of the ground-truth factors in the content block up to an invertible transformation. The information about the factors in the style block is discarded as the model learns "[...] *all and only* information about [the content block]" (Yao et al., 2024c, Definition 2.3).

**Data Generation.** To generate the ground-truth causal factors, we employ the same sampling procedure as for CCRL. Due to its flexible constraints on the underlying factors, the distribution entailed by the SCM and the interventions falls under the method's assumptions, and we pool the observations from the different interventional environments. See Appendix A.2.1 for an in-depth description of the data-generating process. The input to the method consists of 60K observations, whose variables are grouped into the four views shown in Figure 4. The first view is the images ($\tilde{\text{Im}}$) captured by the camera, which depends on all ground-truth factors ($R, G, B, \theta_1, \theta_2$). The second view comprises the combined measurements of the current drawn by the light source ($\tilde{C}$) and the light intensity at the first two sensors ($\tilde{I}_1, \tilde{V}_1, \tilde{I}_2, \tilde{V}_2$); these depend only on the light-source setting ($R, G, B$). The third and fourth views are the measurements of the two angle sensors $\tilde{\theta}_1$ and $\tilde{\theta}_2$. These are simple linear functions (e.g., Figure 5D) of the polarizer positions $\theta_1$ and $\theta_2$, respectively. In line with the experiments in the original paper (Yao et al., 2024c, Section 5.3), during training we rely on the ground-truth content selection of the shared encodings, as opposed to learning it. More details can be found in Appendix A.2.1.

**Results.** In Figure 5, we display the results of applying the multiview CRL method (Yao et al., 2024c) on both the real data and the synthetic ablation described in Section 2. We report the same metric used in the original paper: the $R^2$ score (Wright, 1921) of predicting each ground-truth factor using a non-linear estimator, fit on the encodings returned by the method for each view, and averaged for each pair of views shown in panels A-C of Figure 5. Further details about the metric can be found in Appendix A.2.3.

Success is indicated by simultaneously attaining a high score (near one) for the underlying factors shared by both views (the content block) and a score significantly lower than one for those factors that are not (the style block). In the case where the underlying factors are independent, the score for the style is expected to be near zero, but given the correlation between variables induced by the underlying SCM, this is not the case here. While the method consistently recovers the variables in the content block better than those in the style block, the difference in scores is not large, as is particularly evident for the intersections of view 1 with views 3 and 4, where this difference is barely noticeable (Figure 5 B and C, respectively). This implies that the method fails to separate the information between the content and style variables, which is one of the method's goals (Yao et al., 2024c, Definition 2.3). It is noteworthy that there is no prominent decrease in performance from the synthetic to the noisy real data. A hypothesis for this phenomenon lies in the subtlety of the trace of the polarizer angle information ($\theta_1, \theta_2$) in the image data and the model's failure to learn this pattern in the image encoder. Given the fact that image data is captured

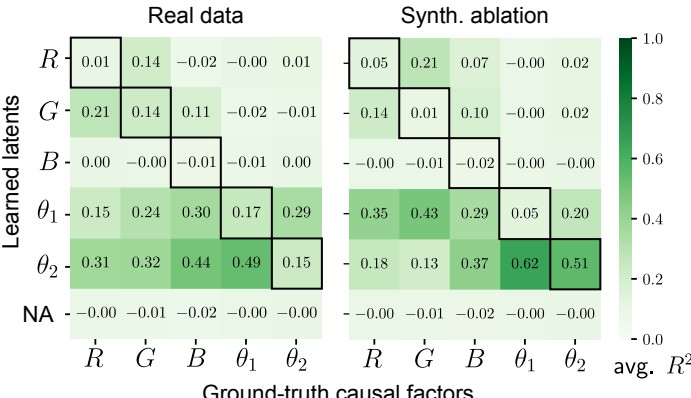

|  | $R^2$ diag ↑ | $R^2$ sep ↓ |
|---|---|---|
| Real | $0.092 \pm 0.056$ | $0.620 \pm 0.091$ |
| Synth. ablation | $0.120 \pm 0.068$ | $0.636 \pm 0.107$ |

Figure 6: Results for the CITRIS method on real and and the synthetic ablation as described in Section 2. **Left:** $R^2$ correlation metric over five random initializations. We report the average and ($\pm$) the standard deviation. The *diag* columns refer to the average value of the diagonal of the $R^2$ correlation matrix (shown on the left) while the *sep* columns denote the maximum off-diagonal value. **Right:** Average $R^2$ correlation matrices for real and synthetic ablation datasets. NA refers to the group of learned latents that are not assigned to any ground-truth causal factor during training (Lippe et al., 2022, Section 3.4). While there is a minor drop in performance from synthetic to the real data, this is negligible and the method fails to recover the ground-truth factors in both cases, as indicated by the low scores along the diagonal and (relatively) high off-diagonal scores, respectively.

in distinct red, green and blue channels, it is unsurprising the ground-truth factors $(R, G, B)$ are more easily learned by the image encoder than the polarizer angles. However, as we show in Appendix B the information of the angles is indeed present in the images, giving us no reason to believe this task to be fundamentally ill-posed.

### 3.3. CRL from Temporal Intervened Sequences

As a third group of CRL approaches, we consider methods that exploit the causal temporal structure of a process evolving in time (Yao et al., 2021; 2022; Lippe et al., 2022; 2023a;b; Lachapelle et al., 2024). To represent this group of methods, we select CITRIS (Lippe et al., 2022), for which well-documented code is publicly available.

**Background.** CITRIS assumes that the underlying causal factors follow a first-order Markov, stationary dynamic Bayesian network (DBN, Dean & Kanazawa, 1989; Murphy, 2002) without instantaneous causal effects (Lippe et al., 2022, Section 2). Furthermore, the method assumes to have access to information about which underlying causal factor has undergone an intervention at each time step (Lippe et al., 2022, Section 3.1). In contrast to most other methods, CITRIS allows for multidimensional causal ground-truth variables. It explicitly models observations as variables with noise in its theoretical assumptions (Lippe et al., 2022, Section 3.1).

**Data Generation.** We generate the ground-truth causal factors by sampling from the temporal causal process described in Appendix A.3.1. As in the experiments of the original paper (Lippe et al., 2022, Appendix C.2), at each time step we randomly intervene on one of the factors (also including the possibility of intervening on none), and record the vector of the intervention targets at each time step. The original paper uses an additional loss metric (the "triplet" loss, Lippe et al., 2022, Section 6.1) for model selection during training, which requires access to the ground-truth factors and requires the collection of an additional dedicated dataset. Since collecting data on a physical system is costly, and assuming access to the ground-truth factors during training is unrealistic, we forego the use of this metric. We use the CITRIS-VAE variant of this method (Lippe et al., 2022, Section 4.1) for all experiments.

As in the original paper (Lippe et al., 2022, Section 6.1), we generate an additional dataset from independently sampled underlying factors to calculate the final metrics: the $R^2$ (Wright, 1921) and Spearman's rank correlation coefficients (Spearman, 1904), which show how well the model can predict the ground-truth factors from the latents learned during training. Please see Appendix A.3.3 for details on these metrics.

**Results.** We present the results for the $R^2$ score in Figure 6; the results for the Spearman's rank coefficient are provided in Appendix A.3.4. Both metrics—employed in the original paper—produce a correlation matrix between

the learned latents and the ground-truth factors. An optimal model displays values close to one on the diagonal and zeros everywhere else (see Appendix A.3.3 for details). The results indicate that CITRIS fails catastrophically in recovering the ground-truth factors, both from the real data and the ablation with synthetic images and measurements. The metrics along the diagonal display a low score, while the off-diagonal values are relatively higher. This suggests that the model does not learn an encoding that separates the ground-truth causal factors nor a correct mapping between latents and causal factors.

CITRIS is a complex method with many sub-modules that interact during training. We hypothesize that a failure in one of these modules, such as the encoder struggling to extract information about one of the latents, can compound and drive down the method's performance. Although CITRIS and its variants have shown favorable performance on several visually complex synthetic datasets (Lippe et al., 2022; 2023a;b), the exact reason for the failure in this setting remains elusive. Further analysis to pinpoint where things go wrong would require an in-depth study and additional ablations on the underlying data-generating process, e.g., by considering a simpler benchmark with only $R, G$, and $B$ as ground-truth factors. We believe this to be beyond the scope of this work.

## 4. Discussion

We have constructed a testbed for causal representation learning using a real and tightly controlled physical system built around a well-understood optical experiment. The data-generating process of this system satisfies the core assumptions of causal representation learning, and the straightforward mixing process is described by simple physical principles. Furthermore, we have access to ground-truth labels of the underlying causal factors. Therefore, the testbed serves as a sanity check where a CRL method that is expected to work on a variety of real-world scenarios should also succeed. A failure on this testbed indicates a potential failure on other, more complex real-world systems. However, the opposite does not hold, as success on this simple and controlled testbed is not guaranteed to carry over to more complex scenarios.

As a first application of our testbed, we evaluated three methods representative of different approaches to causal representation learning: contrastive CRL, multiview CRL, and CRL from temporal intervened sequences. Due to the different setups, assumptions, and goals of each method, we performed our analysis on a method-by-method basis. However, some general patterns emerge. First, all methods failed to attain their goals in recovering—up to their corresponding theoretical constraints—the ground-truth causal factors. Except for a single sub-result of the multiview method, the

failure is catastrophic, and the output of the methods has a very weak or no association with the ground-truth causal factors. To better understand the reasons for this failure, we performed a synthetic ablation on the data by constructing a deterministic simulator to substitute the real-world data-generating process of the light tunnel. However, save for the CCRL method, performance on this simpler synthetic data was not significantly better than on the real data.

To better understand the failure modes of the algorithms, it is useful to draw the distinction between failure due to misspecification and failure due to optimization. To prove identifiability, CRL algorithms assume a data-generating process with different assumptions surrounding distributions, number and type of interventions, and the functional form of mechanisms, among others. Given a generative process that follows these assumptions, methods typically show that optimizing a certain objective—under the assumption of infinite data—recovers the data-generating causal variables up to acceptable indeterminacies such as permutation and element-wise rescaling of variables. Thus, failures in practice can be attributed either to a misspecification of the assumed data-generating process or to issues in the optimization procedure.

To control for misspecification failures, we exactly adhere to each method's assumptions surrounding the data-generating process, leaving only the real-world mixing function as a source of misspecification. As a negative control for its effect, we carry out synthetic ablation experiments, using deterministic simulators as well-specified synthetic surrogates of the real-world mixing function. The CCRL method succeeds in recovering the latent variables from the synthetic ablation data and fails on the real data, providing evidence that the failure mode for this method can be traced back to a misspecification of the real-world mixing function. The same conclusion cannot be readily drawn for the multiview CRL method and CITRIS, as they fail on both the real and synthetic ablation data. We must conclude that their failure likely lies in a combination of both misspecification and optimization failure.

To exclude key potential failures in the optimization process, we check for non-decreasing losses and we perform a large hyperparameter search for all methods (cf. Appendix D). Additionally, we make a substantial effort to exclude bugs and other issues by performing preliminary checks on our pipeline, reproducing the synthetic experiments in the original papers, and consulting the respective authors to advise us where possible. Exhaustively covering the vast space of all factors surrounding optimization, such as alternative model architectures and training regimes, is beyond the scope of this work, which is a sanity check for existing methods as they would be available to a practitioner. For the multiview CRL method and CITRIS, we cannot exclude

that architecture choices, as well as finite sample issues, are not responsible for their failure.

This opaque understanding of failure modes highlights a deeper problem, underscoring the fragility of these methods and their dependence on pre-processing steps and other implementation decisions, such as model architectures, training regimes, and hyperparameter values. This constitutes a significant challenge for reproducibility and, ultimately, the application of these methods to real-world problems.

### 4.1. Outlook

Although we advocate for the use of real-world data, we believe that synthetic data is crucial for the development and validation of new methods. However, there is an inherent conflict of interest if the validation of a new method is limited to synthetic data produced by its authors. Method-agnostic benchmarks have been catalysts of progress in various other fields (LeCun et al., 1998; Deng et al., 2009; Rajpurkar et al., 2016; Runge et al., 2019; Lin et al., 2022), especially if they include or resemble data that we may find in the real world. Thus, we applaud existing efforts in this regard, including the synthetic benchmarks mentioned throughout the manuscript (Lippe et al., 2022; 2023a; Ahmed et al., 2021; von Kügelgen et al., 2021; Liu et al., 2023).

While the community acknowledges that applications of CRL to real-world problems are lacking, the response has largely been to push the theoretical frontier and relax the assumptions needed to establish identifiability results. We argue that, in many regards, the theory in the field is already quite advanced and is, therefore, not the only bottleneck. It is equally important to make a real effort to implement and apply the existing theory in a robust, reproducible, and efficient manner. Without pursuing the engineering endeavor to develop actionable algorithms from the existing theory, we cannot hope to know where the true obstacles to application lie.

Unless we abandon the ultimate goal of applying causal representation learning to real-world problems, we cannot be satisfied with validating our methods on purely synthetic data. Our goal with this physical testbed is to provide researchers in the field with a stepping stone toward more complex real-world scenarios, with the hope of breaking down this hard problem into manageable steps.

The datasets we provide constitute a small fraction of all experiments possible with the light tunnel, which offers additional control inputs not described here that can be used to construct more complex tasks. We are open to suggestions for additional experiments that may prove useful—please reach out to the corresponding authors.

## Acknowledgements

We thank Francesco Locatello for valuable discussions, and Simon Buchholz and Dingling Yao for providing support for their respective model implementations. We also wish to thank the anonymous reviewers for their comments and feedback. S.B. and J.R. received funding for this project from the European Research Council (ERC) Starting Grant CausalEarth under the European Union's Horizon 2020 research and innovation program (Grant Agreement No. 948112). S.B. received support from the German Academic Scholarship Foundation. This work used resources of the Deutsches Klimarechenzentrum (DKRZ) granted by its Scientific Steering Committee (WLA) under project ID 1083.

## Impact Statement

This paper provides a sanity check to validate advances in causal representation learning. Because our contribution pertains to fundamental and theoretical research in machine learning, the potential societal consequences are difficult to ascertain and, in any case, not immediate. Thus, we do not feel any ethical concerns must be specifically highlighted here.

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

# A. Experimental Details

In this section we provide details on all data generation procedures, hyperparameters, training procedures, and minor adjustments made to the original implementations required to reproduce the results of this work. Our general approach was to use each method's existing code as out-of-the-box as possible, and almost all hyperparameters and settings are the same as reported in the original papers. However, slight modifications were inevitable in making the code run for our data, and we thank the authors of the methods for their guidance. All implementations use the PyTorch machine learning library (Paszke et al., 2019). The data used in this work is available through the `lt_crl_benchmark_v1` dataset at `github.com/juangamella/causal-chamber` and the code, including all implementations of the considered methods, is available at `https://github.com/simonbing/CRLSanityCheck`.

**Computational Resources and Compute Time.** All experiments were run on a high-performance cluster with NVIDIA A100 GPUs. The total compute time required to reproduce all experiments in this work is approximately 100 GPU hours.

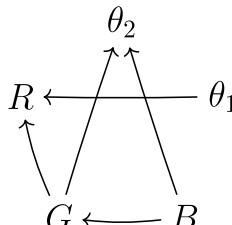

Figure 7: The ground-truth graph for the Contrastive CRL dataset.

## A.1. Contrastive CRL

### A.1.1. DATA GENERATION

We define a linear SCM with additive Gaussian noise between the variables $(R, G, B, \theta_1, \theta_2)$ with causal relations encoded by the graph shown in Figure 7. The independent Gaussian noise variables all have zero mean and a variance sampled from $\mathcal{U}([0.01, 0.02])$. To generate interventional data, we consider interventions that shift the mean of their target by adding a factor $\eta$. We sample the value of the shift $\eta$ from $\mathcal{U}([1, 2])$ for all interventions. Each variable is intervened upon individually and we collect $n = 10000$ samples from each intervention, as well as the observational distribution. The full dataset is then split into train, validation, and test subsets according to the ratios (80/10/10) while ensuring that each subset contains the same fraction of samples from each environment.

### A.1.2. TRAINING DETAILS

For our experiments with the CCRL method, we rely on the code kindly provided by the authors of the original work. As the original implementation only included a shallow convolutional encoder for images, we found it useful to extend the image encoder to a slightly deeper architecture, which we report in Table 1. Otherwise, we use the exact same architecture and implementation as in the original paper. We report the hyperparameters used during training in Table 2.

Our initial experiments revealed it was crucial that the noise terms of the underlying SCM have mean zero. Attempts with a non-zero mean led to the failure of the method.

### A.1.3. METRICS

**Mean Correlation Coefficient.** The Mean Correlation Coeefficent (MCC, Hyvärinen & Morioka, 2016; Khemakhem et al., 2020) measures the pairwise correlation between the learned representation and the ground truth latents. If $\mathbf{C} \in \mathbb{R}^{d \times d}$ is the Pearson correlation matrix between the ground truth variables and the learned embeddings and $\mathfrak{S}_d$ denotes the set of $d$-permutations, the MCC score is formally defined as

$$\text{MCC} := \max_{\pi \in \mathfrak{S}_d} \frac{1}{d} \sum_{j=1}^{d} |\mathbf{C}_{j, \pi(j)}|.$$

**Structural Hamming Distance.** The Structural Hamming Distance (SHD, Tsamardinos et al., 2006) measures how well a method recovers the ground-truth causal graph by comparing the adjacency matrices of the ground-truth graph $\mathbf{A} \in \{0,1\}^{d \times d}$ and the learned graph $\hat{\mathbf{A}} \in \{0,1\}^{d \times d}$, and counting the number of edges that are different. Incorrectly oriented edges count as two errors. We follow the convention that for an adjacency matrix $\mathbf{A}$, a nonzero entry $\mathbf{A}_{i,j} \neq 0$ implies an edge $i \rightarrow j$. Since the Contrastive CRL method outputs a continuous estimate of the adjacency matrix, it must the thresholded to obtain binary entries. We follow the heuristic in the original paper and choose the thresholding that results in the same number of edges as the ground truth $\mathbf{A}$. Naturally, this is not possible when the ground-truth graph is not known. The SHD is then defined as

$$\text{SHD} := \sum_{i,j} |\mathbf{A}_{i,j} - \hat{\mathbf{A}}_i, j|.$$

Table 1: Convolutional encoder architecture for Contrastive CRL.

| Layer | Activation function |
|---|---|
| Conv (c_in = 3, c_out = 64, kernel = 3, stride = 2) | GroupNorm+SiLU |
| Conv (c_in = 64, c_out = 64, kernel = 3, stride = 1) | GroupNorm+SiLU |
| Conv (c_in = 64, c_out = 64, kernel = 3, stride = 2) | GroupNorm+SiLU |
| Conv (c_in = 64, c_out = 64, kernel = 3, stride = 1) | GroupNorm+SiLU |
| Reshape ($1 \times 1 \times (16 \cdot 16 \cdot 64)$) | - |
| Linear (($16 \cdot 16 \cdot 64$) $\times$ 1024) | LayerNorm+SiLU |
| Linear (1024 $\times$ 5) | - |

Table 2: Contrastive CRL hyperparameters.

| Parameter | Value |
|---|---|
| Model latent dimension | 5 |
| Learning rate | 5e-4 |
| Optimizer | Adam (Kingma & Ba, 2015) |
| Epochs | 100 |
| Batch size | 512 |
| Learning rate scheduler | Plateau Scheduler |
| $\kappa$ | 0.1 |
| $\tau_1$ (Buchholz et al., 2023, Eq. 9) | 1e-5 |
| $\tau_2$ (Buchholz et al., 2023, Eq. 9) | 1e-4 |

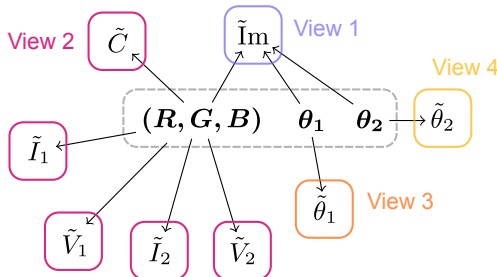

Figure 8: Ground-truth graph relating the latent causal factors, shown in bold print, to the different views employed in the Multiview CRL experiment. The views are disjoint sets of the output variables produced by the tunnel. The graph is a subgraph of the complete causal ground-truth graph for the light tunnel, described in Gamella et al. (2025, Figure 3b).

## A.2. Multiview CRL

### A.2.1. DATA GENERATION

The underlying causal model used in our experiments for the Multiview CRL method is identical to the one used for the Contrastive CRL method described in Appendix A.1.1. In addition to first view given by the images ($\tilde{\text{Im}}$)—which depend on ($R, G, B, \theta_1, \theta_2$)—we consider three additional views. The second view consists of the electrical current drawn by the light source and the infrared and visible light intensities from sensors one and two, i.e., ($\tilde{C}, \tilde{I}_1, \tilde{V}_1, \tilde{I}_2, \tilde{V}_2$). These depend on the underlying causal factors ($R, G, B$). The third and fourth views consist of the angle measurements of the first ($\tilde{\theta}_1$) polarizer and second polarizers, respectively. In turn, they depend on the corresponding polarizer angles ($\theta_1$) and ($\theta_2$). A graphical overview of the data-generating process is shown in Figure 8. These views consist of either real images and sensor measurements collected from the tunnel, or their synthetic and deterministic counterparts produced by the simulator given in Appendix C. To increase the overall sample size, we include observational samples, as well as all interventional samples described in Appendix A.1.1, arriving at a dataset with $n = 60000$ samples. The data is randomly split into train, validation, and test sets according to a 80/10/10 ratio.

### A.2.2. TRAINING DETAILS

We use the publicly available implementation of the Multiview CRL method from https://github.com/CausalLearningAI/multiview-crl. For our experiments, we use the variant of this implementation that relies on BarlowTwins (Zbontar et al., 2021), in accordance with the experiments that use image data in the original paper. The hyperparameter settings used in our experiments are reported in Table 3. The number of training steps is chosen to obtain approximately the same number of epochs given our chosen batch size as the original paper uses for their image data experiments.

Table 3: Multiview CRL hyperparameters.

| Parameter | Value |
| --- | --- |
| Model latent dimension | 5 |
| Learning rate | 1e-4 |
| Optimizer | Adam (Kingma & Ba, 2015) |
| Nr. of training steps | 28550 |
| Batch size | 512 |
| Similarity metric | Cosine similarity |

### A.2.3. METRICS

The theory of the Multiview CRL method states that the model achieves *block-identifiability* (Yao et al., 2024c, Definition 2.3), meaning that the information of the ground-truth variables belonging to the content block of a set of views is perfectly

recovered by the model, while the information of the style variables belonging to this set of views is ignored. The authors measure recovery by looking at the $R^2$ score of an estimator trained to predict the values of the individual ground-truth latents from specific subsets of the learned representation. For each view in a given set, the view-specific encoder returns an encoded latent vector $\hat{\mathbf{z}} \in \mathbb{R}^d$. Let $[d] = (1, ..., d)$ denote the set of integers from one to $d$. Then, a subset of the view-specific encoding vector $\hat{\mathbf{z}}_S, S \subseteq [d]$ is selected as input to the estimator that predicts the learned latent variables. This subset differs between distinct sets of views and corresponds to what the model learns as the *content-block* of that respective set of views. For each latent variable and view in a given set, we record the $R^2$ score (Wright, 1921) of the estimator fit according to the aforementioned procedure, averaging the scores over the encodings obtained by different view-specific encoders in a given set. According to the theory, if the model achieves block identifiability, this metric will be close to one for the variables belonging to the content block in a set of views, while being significantly lower for variables that belong to the style partition.

### A.3. CITRIS

#### A.3.1. DATA GENERATION

We generate the data for the CITRIS experiments by defining a time-evolving process given by the following time-series graph and assignments:

$$
\begin{array}{cccc}
G^t & B^t & \theta_2^t & \theta_1^t \\
\downarrow & \downarrow & \downarrow & \downarrow \\
G^{t+1} & B^{t+1} & \theta_2^{t+1} & \theta_1^{t+1} \\
& \nwarrow & \uparrow & \nearrow \\
& & R^t \longrightarrow R^{t+1}
\end{array}
\qquad
\begin{aligned}
R^{t+1} &\leftarrow f\left(R^t + \epsilon_R\right) \\
G^{t+1} &\leftarrow f\left(G^t + \frac{R^t - G^t}{2} + \epsilon_G\right) \\
B^{t+1} &\leftarrow f\left(B^t + \frac{G^t - B^t}{4} + \epsilon_B\right) \\
\theta_1^{t+1} &\leftarrow g\left(\theta_1^t + \epsilon_1\right) \\
\theta_2^{t+1} &\leftarrow g\left(\theta_2^t + S(R^t, B^t)\left(\frac{\theta_1^t - \theta_2^t}{4}\right)\epsilon_2\right),
\end{aligned}
$$

where the random innovations are sampled as $\epsilon_R, \epsilon_G, \epsilon_B \overset{\text{i.i.d.}}{\sim} \text{Unif}[-50, 50]$, $\epsilon_1 \overset{\text{i.i.d.}}{\sim} \text{Unif}[-10, 10]$, and $\epsilon_1 \overset{\text{i.i.d.}}{\sim} \text{Unif}[-5, 5]$. Furthermore, the functions $S, f$ and $g$ are defined as

$$
S(R^t, B^t) := \begin{cases} 1 & \text{if } B^t > R^t \\ -1 & \text{if } B^t \leq R^t \end{cases}, \quad f(x) := \begin{cases} -x & \text{if } x < 0 \\ 510 - x & \text{if } \geq 255 \end{cases}, \quad \text{and } g(x) := \begin{cases} -180 - x & \text{if } x < -90 \\ 180 - x & \text{if } x \geq 90 \end{cases}.
$$

The purpose of $f$ and $g$ is to ensure that the values of $R^t, \ldots, \theta_2^t$ remain within the bounds of the control inputs for the light tunnel ($[0, 255]$ for $R, G, B$ and $[-90, 90]$ for $\theta_1, \theta_2$).

Intuitively, the process can be understood as follows: the variables $R$ and $\theta_1$ follow random walks. $G^t + 1$ tries to approach $R^t$ in each step, and $B^t + 1$ approaches $G^t$ in the same fashion. However, the innovations $\epsilon_R, \epsilon_G$, and $\epsilon_B$ prevent these variables from converging to their targets. Finally, $\theta_2$ tries to exploit the color-shifting effect of the polarizers to maximize the ratio of the blue to red pixels in the resulting images. It does so by approaching $\theta_1^t$—aligning the polarizer axes—if $B^t > R^t$, and receding if $R^t \geq R^t$. The behavior is regulated by the function $S(\cdot)$ above. At each time step, we perform no intervention with probability 0.3, or we select an intervention target at random from $\{R^t, G^t, B^t, \theta_1^t, \theta_2^t\}$, setting it to a value sampled uniformly at random from its valid range. We encode the intervention target at each time step in a one-hot vector, with a vector of zeroes indicating that no variable is intervened upon. A Python implementation of the process is provided in `github.com/juangamella/causal-chamber/blob/main/datasets/lt_crl_benchmark_v1/generators/citris_1.py`.

We collect a sequence of $n = 100000$ samples from this process and use the first $n_{\text{train}} = 80000$ samples as the training dataset and the following $n_{\text{val}} = 10000$ and $n_{\text{test}} = 10000$ as validation and test datasets. Following the experiments in the original paper, to evaluate the final metrics we collect an additional dataset with $n = 1000$ samples, where the latent variables are sampled uniformly at random from their valid range.

A.3.2. TRAINING DETAILS

For the CITRIS experiments we rely on the code that is publicly available at
https://github.com/phlippe/CITRIS. The hyperparameter settings we used in our experiments are
shown in Table 4. As we mention in the main text, we opted not to use the method's triplet loss for model selection, as this
relies on accessing ground-truth latent values during training, which we find highly unrealistic. We replaced this validation
loss with the same loss function used during training. Calculating the correlation metrics at intermediate steps during
training does not suggest that this leads to a suboptimal model selection in terms of the final metric.

During initial experiments, we found that a number of different choices did not result in any improvement of the final results.
We tried using the auto-regressive prior that does not rely on normalizing flows (as opposed to the one that does), setting the
latent dimension of the model to match that of the ground truth, fixing the assignment of the model latents to the causal
variables instead of learning it, as well as training for 500 instead of 250 epochs. None of these choices resulted in an
improved final score.

Table 4: CITRIS hyperparameters.

| Parameter | Value |
| --- | --- |
| Model latent dimension | 16 |
| Learning rate | 1e-4 |
| Optimizer | Adam (Kingma & Ba, 2015) |
| Epochs | 250 |
| Batch size | 512 |
| Learning rate scheduler | Cosine Warmup (25 steps) |

A.3.3. METRICS

Similar to the MCC score, the authors of CITRIS report the correlation between the learned representation and the ground-
truth variables. However, since CITRIS allows for multiple learned latents to describe a single ground-truth variable, their
model first learns a mapping that assigns latent dimensions to a ground-truth factor. To compute the final metrics, an MLP
is trained to predict each ground-truth variable from the learned groupings of model latents. Each ground-truth factor is
predicted from each respective grouping, resulting in a matrix. The authors then report the $R^2$ (Wright, 1921) as well as
Spearman's rank correlation coefficient (Spearman, 1904) for this matrix. As a summary metric, the diagonal of the resulting
matrices is examined, considering its average value, which should be close to one in the optimal case, as well as the maximal
off-diagonal value, which is expected to be close to zero.

A.3.4. ADDITIONAL FIGURES

We present additional results of the CITRIS experiments (cf. Section 3.3) using Spearman's rank correlation coefficient in
Figure 9. We see that this metric echoes the findings presented in Figure 6.

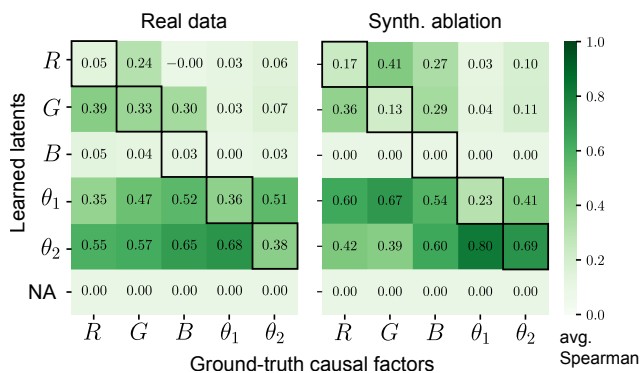

Figure 9: Results for the CITRIS method on real and and the synthetic ablation as described in Section 2. **Left:** Spearman's rank correlation coefficient over five random initializations. We report the average and ($\pm$) the standard deviation. The *diag* columns refer to the average value of the diagonal of the Spearman correlation matrix while the *sep* columns denote the maximum off-diagonal value. **Right:** Average Spearman's rank correlation matrices for real and synthetic ablation datasets. NA refers to the group of learned latents that are not assigned to any ground-truth causal factor during training (Lippe et al., 2022, Section 3.4). Analogously to the $R^2$, the results here show that the method fails fails to recover the ground-truth factors both for real and synthetic data, as indicated by the low scores along the diagonal and (relatively) high off-diagonal scores, respectively.

|                  | Spearman diag $\uparrow$ | Spearman sep $\downarrow$ |
|------------------|------------------|------------------|
| Real             | $0.230 \pm 0.058$ | $0.801 \pm 0.052$ |
| Synth. ablation  | $0.245 \pm 0.055$ | $0.809 \pm 0.072$ |

## B. Supervised Sanity Check

To ensure that the information of the latent variables $(R, G, B, \theta_1, \theta_2)$ can indeed be recovered from the image data, we train a small MLP (architecture cf. Table 6) in a supervised fashion to predict the ground-truth causal factors from images on a held-out test set. We train on a subset of 5000 samples and test on 500 samples. Our results (cf. Table 5) indicate that the latent variables do indeed leave a detectable trace in the data. We stress that this is merely a sanity check to exclude that our data is collected in a contrived way. The near-perfect performance of this supervised model should not be compared with the unsupervised methods we investigate in the main paper.

Table 5: Test score of a supervised model trained to predict latents from images for the respective datasets.

|         | CCRL & Multiview CRL Dataset | CITRIS Dataset |
|---------|------------------|------------------|
| $R^2$   | 0.976            | 0.914            |

Table 6: Supervised MLP network architecture.

| Layer | Activation function |
|-------|---------------------|
| Linear $((3 \cdot 64 \cdot 64) \times 64)$ | LeakyReLU(0.01) |
| Linear $(64 \times 256)$ | LeakyReLU(0.01) |
| Linear $(256 \times 256)$ | LeakyReLU(0.01) |
| Linear $(256 \times 64)$ | LeakyReLU(0.01) |
| Linear $(64 \times 5)$ | - |

## C. Deterministic Simulator of the Light Tunnel

In this section, we describe the deterministic simulator used for the synthetic ablations of the light-tunnel data in Section 3. The simulator consists of two components: a mechanistic model of the sensors (Appendix C.1), derived from first principles and the technical datasheets of the sensors, and a neural simulator to produce synthetic images (Appendix C.2). An implementation of both components, using Numpy and Pytorch, is provided by the `causalchamber` package; see simulators `lt.Deterministic` and `lt.DecoderSimple` in the simulator index for the source code and Jupyter notebooks with examples on how to run them.

For completeness, we provide a diagram of the light tunnel in Figure 10, including additional variables used by the simulator that are not shown in Figure 1. A detailed description of all involved variables is given in Table 7, and the simulator parameters are described in Table 8.

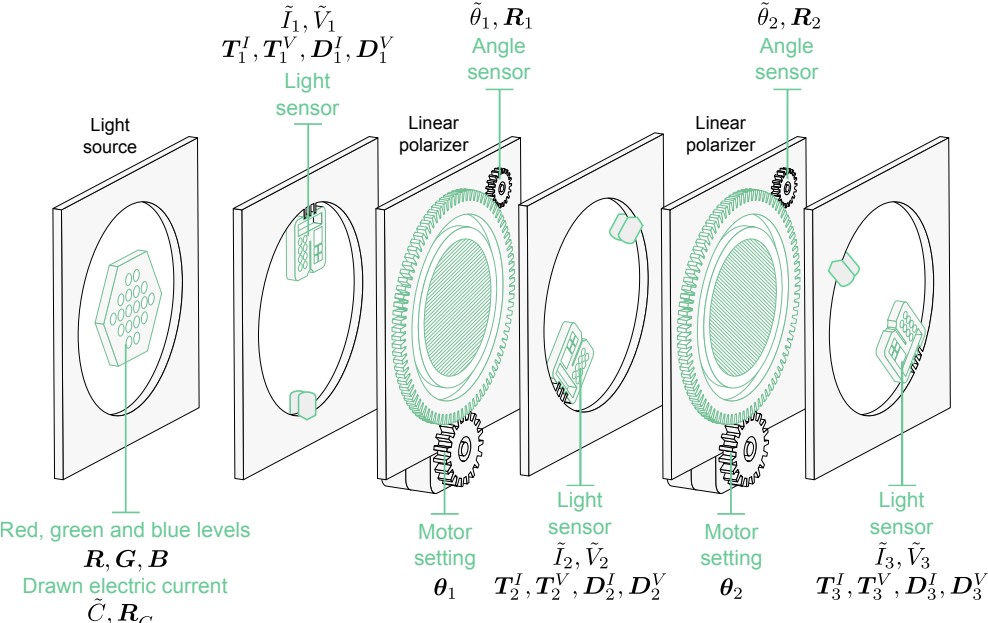

Figure 10: Diagram of the light tunnel with the variables involved in the deterministic simulator, where bold symbols correspond to the inputs to the simulator, and its outputs are denoted by a tilde. The inputs include variables not shown in Figure 1, which control the behavior of the current sensor ($R_C$), the light-intensity sensors ($D_*^I, D_*^V, T_*^I, T_*^V$), and the angle sensors ($R_1, R_2$). All variables are described in detail in Table 7 below, and a complete description of the light tunnel can be found in Gamella et al. (2025).

Table 7: Description of the light tunnel variables involved in the deterministic simulator, including their symbol, value range, and the corresponding column name in the dataset files.

| Variable | Type | Data column | Description |
|---|---|---|---|
| $\tilde{I}_1, \tilde{I}_2, \tilde{I}_3 \in \{0, \dots, 2^{16} - 1\}$ | Output | `ir_1` `ir_2` `ir_3` | The uncalibrated measurement of the light-intensity sensors in the infrared part of the spectrum, placed before ($\tilde{I}_1$), between ($\tilde{I}_2$), and after ($\tilde{I}_3$) the polarizers, in reference to the light source. |
| $\tilde{V}_1, \tilde{V}_2, \tilde{V}_3 \in \{0, \dots, 2^{16} - 1\}$ | Output | `vis_1` `vis_2` `vis_3` | The uncalibrated measurement of the light-intensity sensors in the visible part of the spectrum, placed before ($\tilde{V}_1$), between ($\tilde{V}_2$), and after ($\tilde{V}_3$) the polarizers, about the light source. |

Table 7: Description of the light tunnel variables involved in the deterministic simulator, including their symbol, value range, and the corresponding column name in the dataset files.

| Variable | Type | Data column | Description |
|---|---|---|---|
| $\tilde{C} \in [0, 1023]$ | Output | current | The uncalibrated measurement of the electric current drawn by the light source. The calibrated measurement (in amperes) is given by $$\tilde{C} \times \frac{R_C}{1023 \times 5} \times 2.5.$$ |
| $\tilde{\theta}_1, \tilde{\theta}_2 \in [0, 1023]$ | Output | angle_1 angle_2 | The position of the polarizers is encoded into a voltage using a rotary potentiometer, which is then read by the control computer to produce the measurements $\tilde{\theta}_1$ and $\tilde{\theta}_2$. Given these measurements, the calibrated angle measurement is given as $$(\tilde{\theta}_j - Z_j) \times \frac{720}{1023} \times \frac{\text{vref}(R_j)}{5} \text{ degrees,}$$ where $R_j$ is the reference voltage of the corresponding sensor, and $Z_1 = 507, Z_2 = 512$ are the readings at angles $\theta_1 = \theta_2 = 0$ and reference voltages $R_1 = R_2 = 5$. |
| $R, G, B \in \{0, \ldots, 255\}$ | Input | red green blue | The brightness setting of the red, green, and blue LEDs on the tunnel's light source. Higher values correspond to higher brightness. |
| $\theta_1, \theta_2 \in \{-180, -179.9, \ldots, 180\}$ | Input | pol_1 pol_2 | Position of the tunnel's polarizers, in degrees. |
| $D_1^I, D_2^I, D_3^I \in \{0, 1, 2\}$ | Input | diode_ir_1 diode_ir_2 diode_ir_3 | The size of the photodiode used by the light sensors to take infrared measurements, corresponding to the small $(D_j^I = 0)$, medium $(D_j^I = 1)$ and large $(D_j^I = 2)$ photodiodes onboard. |
| $D_1^V, D_2^V, D_3^V \in \{0, 1\}$ | Input | diode_vis_1 diode_vis_2 diode_vis_3 | The size of the photodiode used by the light sensors to take visible-light measurements, corresponding to the small $(D_j^V = 0)$, medium $(D_j^V = 1)$ and large $(D_j^V = 2)$ photodiodes onboard. |
| $T_1^I, T_2^I, T_3^I, T_1^V, T_2^V, T_3^V \in \{0, 1, 2, 3\}$ | Input | t_ir_1/2/3 t_vis_1/2/3 | The exposure time of the photodiode during a light-intensity measurement. Higher values correspond to longer exposure times. |
| $R_C, R_1, R_2 \in \{1.1, 2.56, 5\}$ | Input | v_c v_angle_1 v_angle_2 | The reference voltage, in volts, of the sensors used to measure the current ($\tilde{C}$) and polarizer angles ($\tilde{\theta}_1, \tilde{\theta}_2$), respectively. |

### C.1. Simulating the Sensor Measurements

We derive a mechanistic model from first principles and the details provided in the technical datasheets of the involved components. The datasheets are provided at github.com/juangamella/causal-chamber/tree/main/hardware/datasheets. The expressions for the model, with color-coded variables to differentiate between the inputs, outputs, and parameters of the simulator, are given below. A comparison of the resulting synthetic measurements and the true

measurements collected from the light tunnel are provided in Figure 11.

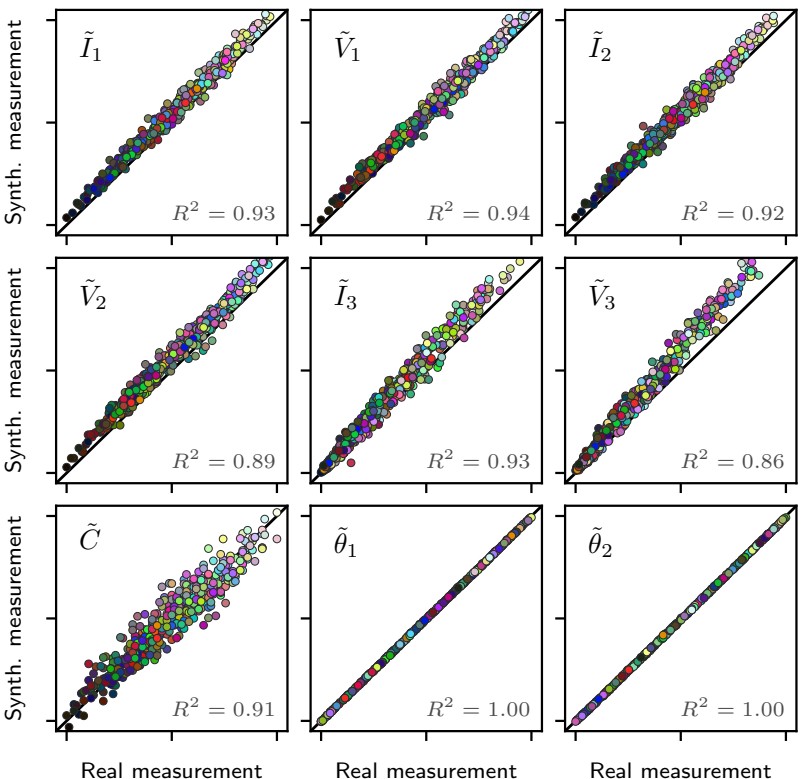

Figure 11: Comparison between the real measurements (x-axis) and the corresponding synthetic measurements (y-axis) produced by the deterministic simulator for the same inputs. Each dot, corresponding to a real/synthetic pair, is colored according to the light-source setting $(R, G, B)$. The diagonal black line corresponds to the deterministic output of the simulator, and we report the $R^2$ score between the real and synthetic measurements.

**Light-Intensity Sensors.** We model the sensor response as a linear combination of the intensity of each light-source color $(R, G, B)$, with coefficients encoded in the matrix $S$. To account for the decrease in intensity as we move away from the light source, we apply the inverse-square law using the distances $d_1, d_2, d_3$ of each sensor to the light source. The effect of the polarizer angles $(\theta_1, \theta_2)$ on the third sensor $(\tilde{I}_3, \tilde{V}_3)$ is modeled through Malus' law (Collett, 2005). Following the specifications of the sensor[1], we model the effect of the photodiode size $(D_i^I, D_i^V)$ and exposure time $(T_i^I, T_i^V)$ as scaling the sensor response by an exponential factor.

---

[1]See pages 42/43 of the Si115x datasheet.

$$\begin{pmatrix} \tilde{I}_1 \\ \tilde{V}_1 \end{pmatrix} := \begin{pmatrix} 2^{D_1^I + T_1^I} & 0 \\ 0 & 2^{D_1^V + D_1^T} \end{pmatrix} S \begin{pmatrix} R \\ G \\ B \end{pmatrix}$$

$$\begin{pmatrix} \tilde{I}_2 \\ \tilde{V}_2 \end{pmatrix} := \begin{pmatrix} 2^{D_2^I + T_2^I} & 0 \\ 0 & 2^{D_2^V + T_2^V} \end{pmatrix} S \left(\frac{d_1}{d_2}\right)^2 T_s \begin{pmatrix} R \\ G \\ B \end{pmatrix}$$

$$\begin{pmatrix} \tilde{I}_3 \\ \tilde{V}_3 \end{pmatrix} := \begin{pmatrix} 2^{D_3^I + T_3^I} & 0 \\ 0 & 2^{D_3^V + T_3^V} \end{pmatrix} S \left(\frac{d_1}{d_3}\right)^2 \operatorname{diag}\left((T_p - T_c)\cos^2(\theta_1 - \theta_2) + T_c\right) \begin{pmatrix} R \\ G \\ B \end{pmatrix}$$

**Current Sensor.** The drawn current $\tilde{C}$ is modeled as a linear combination of the brightness setting $(R, G, B)$ of each light-source color, encoded in the matrix of coefficients $Q$. We include an intercept $C_0$ for the base current drawn even when the light source is turned off ($R = G = B = 0$) and account for the scaling induced by the reference voltage $R_C$.

$$\tilde{C} := \left[ Q \begin{pmatrix} R \\ G \\ B \end{pmatrix} + C_0 \right] \frac{5}{R_C}$$

**Angle Sensor.** The relationship between the polarizer positions $(\theta_1, \theta_2)$ and the voltage measured by the angle sensors $(\tilde{\theta}_1, \tilde{\theta}_2)$ is also affine, and is encoded in the coefficient $A$ and the reference points $a_1, a_2$. As for the current sensor, we account for the scaling induced by the reference voltages $R_1, R_2$. To model the saturation of the sensors at lower reference voltages (see Gamella et al., 2025, Supplementary Figure 9), we take the minimum between the synthetic measurement and the maximum possible value (1023).

$$\tilde{\theta}_1 := \min\left\{ 1023, (A\theta_1 + a_1)\frac{5}{R_1} \right\}, \quad \tilde{\theta}_2 := \min\left\{ 1023, (A\theta_2 + a_2)\frac{5}{R_2} \right\}$$

The parameters of the simulator, displayed in orange in the above equations, are described in Table 8 below. For the synthetic observations used in the experiments of Section 3, the parameters are either taken from the datasheets of the corresponding sensors or fit to a separate calibration dataset. See the tutorial notebook for the actual values and an example of running the Python implementation of the simulator.

Table 8: Description of the parameters of the simulator for the sensor measurements, including their range and the name they take in the Python implementation (see simulator `lt.Deterministic`).

| Parameter | Name in code | Description |
| --- | --- | --- |
| $S \in \mathbb{R}_+^{2\times 3}$ | S | Encodes the linear response of the photodiodes to the light of different colors. The first row $S_{1,i=1,2,3}$ gives the response of the smallest IR photodiode ($D_*^I = 0$) to the red, green, and blue light; the second row corresponds to the smallest visible photodiode ($D_*^V = 0$). |
| $d_1, d_2, d_3 \in \mathbb{R}$ | d1, d2, d3 | The distance, in millimeters, from the tunnel's light source to the first ($d_1$), second ($d_2$) and third ($d_3$) light sensors. |

Table 8: Description of the parameters of the simulator for the sensor measurements, including their range and the name they take in the Python implementation (see simulator `lt.Deterministic`).

| Parameter | Name in code | Description |
|---|---|---|
| $T_s \in [0,1]^3$ | `Ts` | The transmission rate of the first polarizer for each color wavelength. |
| $T_p, T_c \in [0,1]^3$ | `Tp, Tc` | The joint transmission rates of the two polarizers for each color wavelength when their polarization axes are parallel ($T_p$) or orthogonal ($T_c$) to each other. |
| $Q \in \mathbb{R}_+^{1\times 3}$ | `Q` | Encodes the linear response of the current sensor to the brightness setting of each color. |
| $C_0 \in \mathbb{R}_+$ | `c0` | The current drawn by the light source when it is turned off, i.e., $R = G = B = 0$. The value corresponds to the uncalibrated measurement when $R_C = 5$. |
| $A \in \mathbb{R}_+$ | `A` | The linear coefficient relating the change in polarizer position to the change in the voltage reading of the angle sensor, when the reference voltage is $R_j = 5$ for $j = 1$ or 2, respectively. |
| $a_1, a_2 \in \mathbb{R}_+$ | `a1, a2` | The zero point of the angle sensor, i.e., the voltage reading when the polarizer positions are $\theta_1 = 0$ and $\theta_2 = 0$, respectively, and the reference voltages are $R_1 = R_2 = 5$. |

### C.2. Simulating the Images

To simulate the images produced by the light tunnel, we train a simple neural network (Table 9) to generate synthetic images given the inputs $R, G, B, \theta_1$ and $\theta_2$. A comparison of the real and synthetic images is provided in Figure 12. A Python implementation with a pre-trained model is available through the `causalchamber` package (simulator `lt.DecoderSimple`); see the tutorial notebook for example code.

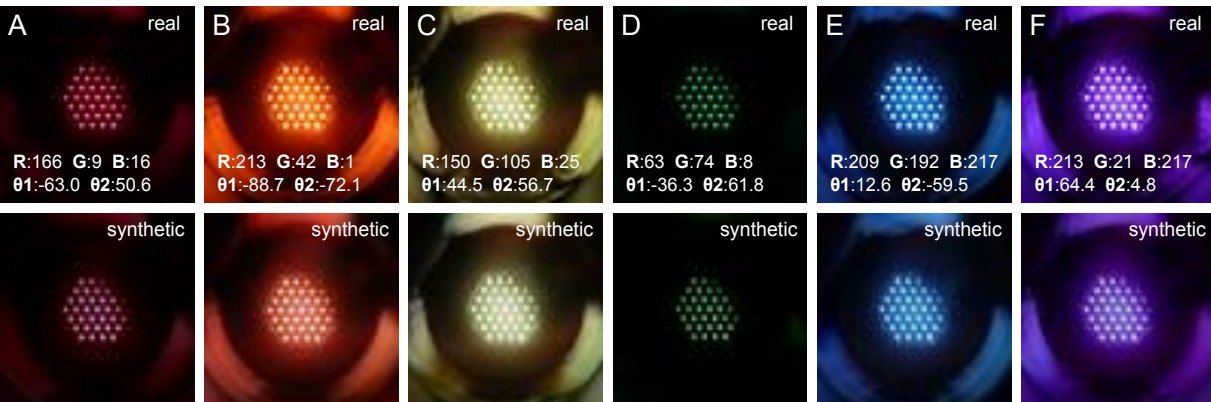

Figure 12: Comparison between the real and synthetic images produced by the deterministic simulator. The simulator does not capture the noise present in the images, and smoothens out some of the finer details produced by the reflection of light on the polarizer frames, which form the ring around the center of the image.

Table 9: Architecture and training parameters of the neural image simulator. The network consists of a stack of 4 fully connected linear layers with ReLu activations. The training loss is the mean-squared error over the pixel intensities. The network is trained on 108900 images from the `lt_camera_v1` and `lt_camera_walks_v1` datasets, which can be found at `github.com/juangamella/causal-chamber`. The code to reproduce the training procedure can be found in this notebook.

| Layers | Parameter | Value |
|---|---|---|
| Linear (5) | Optimizier | Adam (Kingma & Ba, 2015) |
| Linear ($64 \times 64$) | Learning rate | $10^{-3}$ |
| Linear ($64 \times 64$) | Weight decay | $10^{-5}$ |
| Linear ($64 \times 64 \times 3$) | Epochs | 100 |
| | Batch size | 4096 |

## D. Hyperparameter Search

Here we present tables of the results of hyperparameter search experiments for CCRL, multiview CRL and CITRIS. For all models we see that performance is not particularly sensitive to the choice of hyperparameters.

Table 10: Average $R^2$ (3 random seeds) for different hyperparameter settings of the Contrastive CRL method.

| $\tau_2$ | $\tau_1$ | lr 1e−4 0.01 | 0.1 | 1.0 | 5e−4 0.01 | 0.1 | 1.0 |
|---|---|---|---|---|---|---|---|
| 1e−5 | 1e−6 | $0.303 \pm 0.070$ | $0.294 \pm 0.021$ | $0.333 \pm 0.056$ | $0.240 \pm 0.038$ | $0.182 \pm 0.092$ | $0.330 \pm 0.052$ |
| | 1e−5 | $0.324 \pm 0.063$ | $0.285 \pm 0.067$ | $0.312 \pm 0.081$ | $0.278 \pm 0.143$ | $0.296 \pm 0.032$ | $0.322 \pm 0.034$ |
| | 1e−4 | $0.327 \pm 0.040$ | $0.246 \pm 0.063$ | $0.360 \pm 0.042$ | $0.262 \pm 0.066$ | $0.194 \pm 0.088$ | $0.357 \pm 0.028$ |
| 1e−4 | 1e−6 | $0.223 \pm 0.044$ | $0.240 \pm 0.116$ | $0.354 \pm 0.046$ | $0.336 \pm 0.052$ | $0.249 \pm 0.034$ | $0.330 \pm 0.016$ |
| | 1e−5 | $0.307 \pm 0.087$ | $0.261 \pm 0.033$ | $0.341 \pm 0.050$ | $0.278 \pm 0.046$ | $0.287 \pm 0.019$ | $0.327 \pm 0.064$ |
| | 1e−4 | $0.302 \pm 0.071$ | $0.267 \pm 0.033$ | $0.269 \pm 0.037$ | $0.262 \pm 0.075$ | $0.230 \pm 0.070$ | $0.337 \pm 0.038$ |
| 1e−3 | 1e−6 | $0.328 \pm 0.091$ | $0.275 \pm 0.025$ | $0.311 \pm 0.088$ | $0.310 \pm 0.040$ | $0.295 \pm 0.051$ | $0.268 \pm 0.056$ |
| | 1e−5 | $0.261 \pm 0.045$ | $0.221 \pm 0.094$ | $0.345 \pm 0.038$ | $0.337 \pm 0.071$ | $0.313 \pm 0.116$ | $0.283 \pm 0.059$ |
| | 1e−4 | $0.322 \pm 0.046$ | $0.252 \pm 0.016$ | $0.290 \pm 0.065$ | $0.243 \pm 0.084$ | $0.247 \pm 0.027$ | $0.305 \pm 0.095$ |

Table 11: Average SHD (3 random seeds) for different hyperparameter settings of the Contrastive CRL method.

| $\tau_2$ | $\tau_1$ | lr 1e−4 0.01 | 0.1 | 1.0 | 5e−4 0.01 | 0.1 | 1.0 |
|---|---|---|---|---|---|---|---|
| 1e−5 | 1e−6 | $7.333 \pm 1.155$ | $6.667 \pm 1.155$ | $8.000 \pm 2.000$ | $7.333 \pm 1.155$ | $6.667 \pm 1.155$ | $8.667 \pm 2.309$ |
| | 1e−5 | $7.333 \pm 1.155$ | $6.667 \pm 1.155$ | $8.667 \pm 2.309$ | $7.333 \pm 3.055$ | $7.333 \pm 1.155$ | $8.000 \pm 2.000$ |
| | 1e−4 | $7.333 \pm 1.155$ | $6.000 \pm 2.000$ | $7.333 \pm 3.055$ | $6.667 \pm 1.155$ | $6.667 \pm 1.155$ | $8.000 \pm 2.000$ |
| 1e−4 | 1e−6 | $7.333 \pm 1.155$ | $6.667 \pm 1.155$ | $8.667 \pm 2.309$ | $7.333 \pm 2.309$ | $6.000 \pm 2.000$ | $8.000 \pm 3.464$ |
| | 1e−5 | $7.333 \pm 3.055$ | $6.667 \pm 1.155$ | $8.667 \pm 2.309$ | $8.000 \pm 2.000$ | $6.667 \pm 1.155$ | $8.667 \pm 2.309$ |
| | 1e−4 | $7.333 \pm 1.155$ | $7.333 \pm 1.155$ | $8.000 \pm 3.464$ | $6.000 \pm 2.000$ | $6.667 \pm 1.155$ | $8.667 \pm 2.309$ |
| 1e−3 | 1e−6 | $7.333 \pm 1.155$ | $8.000 \pm 2.000$ | $7.333 \pm 1.155$ | $7.333 \pm 1.155$ | $8.000 \pm 3.464$ | $6.667 \pm 1.155$ |
| | 1e−5 | $6.667 \pm 1.155$ | $7.333 \pm 3.055$ | $8.667 \pm 1.155$ | $6.667 \pm 1.155$ | $7.333 \pm 1.155$ | $8.667 \pm 2.309$ |
| | 1e−4 | $6.667 \pm 1.155$ | $8.000 \pm 0.000$ | $7.333 \pm 1.155$ | $7.333 \pm 1.155$ | $8.667 \pm 1.155$ | $7.333 \pm 1.155$ |

Table 12: Average $R^2$ (3 random seeds) for the Multiview CRL hyperparameter search.

| $\tau$ | batch size | lr | view 1 (ĩm) vs. view 2 ($\tilde{C}, \tilde{I}_1, \tilde{V}_1, \tilde{I}_2, \tilde{V}_2$) content↑ R | content↑ G | content↑ B | style↓ $\theta_1$ | style↓ $\theta_2$ | view 1 (ĩm) vs. view 3 ($\tilde{\theta}_1$) style↓ R | style↓ G | style↓ B | content↑ $\theta_1$ | style↓ $\theta_2$ | view 1 (ĩm) vs. view 4 ($\tilde{\theta}_2$) style↓ R | style↓ G | style↓ B | style↓ $\theta_1$ | content↑ $\theta_2$ |
|---|---|---|---|---|---|---|---|---|---|---|---|---|---|---|---|---|---|
| 0.1 | 256 | 1e − 5 | 0.782 ± 0.004 | 0.663 ± 0.005 | 0.855 ± 0.004 | 0.325 ± 0.009 | 0.604 ± 0.010 | 0.029 ± 0.020 | 0.027 ± 0.015 | 0.058 ± 0.027 | 0.354 ± 0.177 | 0.026 ± 0.022 | 0.057 ± 0.062 | 0.156 ± 0.119 | 0.166 ± 0.058 | 0.048 ± 0.033 | 0.380 ± 0.040 |
|  |  | 1e − 4 | 0.806 ± 0.002 | 0.704 ± 0.003 | 0.881 ± 0.002 | 0.301 ± 0.001 | 0.623 ± 0.002 | 0.073 ± 0.038 | 0.114 ± 0.047 | 0.186 ± 0.110 | 0.406 ± 0.123 | 0.130 ± 0.075 | 0.286 ± 0.081 | 0.046 ± 0.023 | 0.103 ± 0.053 | 0.131 ± 0.029 | 0.360 ± 0.112 |
|  |  | 5e − 4 | 0.810 ± 0.002 | 0.700 ± 0.002 | 0.880 ± 0.001 | 0.304 ± 0.005 | 0.616 ± 0.008 | 0.189 ± 0.089 | 0.129 ± 0.113 | 0.161 ± 0.065 | 0.120 ± 0.024 | 0.096 ± 0.051 | 0.094 ± 0.080 | 0.053 ± 0.024 | 0.159 ± 0.089 | 0.048 ± 0.064 | 0.354 ± 0.182 |
|  | 512 | 1e − 5 | 0.768 ± 0.002 | 0.658 ± 0.007 | 0.846 ± 0.006 | 0.307 ± 0.018 | 0.592 ± 0.006 | 0.067 ± 0.066 | 0.022 ± 0.017 | 0.035 ± 0.017 | 0.360 ± 0.130 | 0.012 ± 0.009 | 0.097 ± 0.049 | 0.220 ± 0.145 | 0.231 ± 0.160 | 0.037 ± 0.032 | 0.432 ± 0.080 |
|  |  | 1e − 4 | 0.809 ± 0.002 | 0.705 ± 0.001 | 0.882 ± 0.002 | 0.322 ± 0.015 | 0.643 ± 0.002 | 0.229 ± 0.162 | 0.156 ± 0.181 | 0.127 ± 0.060 | 0.383 ± 0.115 | 0.105 ± 0.079 | 0.164 ± 0.079 | 0.129 ± 0.072 | 0.160 ± 0.087 | 0.064 ± 0.039 | 0.378 ± 0.136 |
|  |  | 5e − 4 | 0.809 ± 0.004 | 0.702 ± 0.002 | 0.879 ± 0.001 | 0.302 ± 0.009 | 0.631 ± 0.006 | 0.139 ± 0.050 | 0.095 ± 0.040 | 0.215 ± 0.161 | 0.331 ± 0.175 | 0.118 ± 0.076 | 0.158 ± 0.028 | 0.033 ± 0.041 | 0.196 ± 0.174 | 0.106 ± 0.024 | 0.425 ± 0.075 |
|  | 1024 | 1e − 5 | 0.750 ± 0.004 | 0.644 ± 0.006 | 0.834 ± 0.003 | 0.292 ± 0.013 | 0.573 ± 0.005 | 0.103 ± 0.081 | 0.015 ± 0.004 | 0.044 ± 0.028 | 0.348 ± 0.143 | 0.023 ± 0.014 | 0.031 ± 0.020 | 0.138 ± 0.106 | 0.085 ± 0.051 | 0.037 ± 0.012 | 0.368 ± 0.039 |
|  |  | 1e − 4 | 0.811 ± 0.002 | 0.699 ± 0.003 | 0.880 ± 0.000 | 0.345 ± 0.017 | 0.632 ± 0.008 | 0.094 ± 0.042 | 0.119 ± 0.027 | 0.160 ± 0.073 | 0.303 ± 0.132 | 0.118 ± 0.046 | 0.216 ± 0.178 | 0.132 ± 0.032 | 0.076 ± 0.017 | 0.059 ± 0.075 | 0.471 ± 0.049 |
|  |  | 5e − 4 | 0.812 ± 0.002 | 0.710 ± 0.003 | 0.881 ± 0.000 | 0.315 ± 0.010 | 0.646 ± 0.010 | 0.142 ± 0.058 | 0.089 ± 0.039 | 0.170 ± 0.056 | 0.372 ± 0.123 | 0.098 ± 0.018 | 0.139 ± 0.059 | 0.187 ± 0.077 | 0.299 ± 0.014 | 0.048 ± 0.024 | 0.223 ± 0.095 |
| 1.0 | 256 | 1e − 5 | 0.802 ± 0.001 | 0.698 ± 0.001 | 0.879 ± 0.001 | 0.319 ± 0.004 | 0.632 ± 0.001 | 0.096 ± 0.088 | 0.113 ± 0.065 | 0.121 ± 0.016 | 0.390 ± 0.071 | 0.094 ± 0.022 | 0.049 ± 0.018 | 0.072 ± 0.086 | 0.124 ± 0.125 | 0.032 ± 0.011 | 0.357 ± 0.028 |
|  |  | 1e − 4 | 0.749 ± 0.003 | 0.697 ± 0.003 | 0.855 ± 0.003 | 0.220 ± 0.006 | 0.613 ± 0.004 | 0.213 ± 0.034 | 0.254 ± 0.161 | 0.315 ± 0.171 | 0.393 ± 0.121 | 0.245 ± 0.132 | 0.212 ± 0.029 | 0.303 ± 0.089 | 0.365 ± 0.170 | 0.043 ± 0.015 | 0.525 ± 0.086 |
|  |  | 5e − 4 | 0.754 ± 0.012 | 0.700 ± 0.002 | 0.856 ± 0.005 | 0.239 ± 0.005 | 0.619 ± 0.005 | 0.152 ± 0.082 | 0.091 ± 0.047 | 0.081 ± 0.050 | 0.286 ± 0.110 | 0.078 ± 0.035 | 0.064 ± 0.029 | 0.197 ± 0.105 | 0.357 ± 0.212 | 0.017 ± 0.010 | 0.348 ± 0.132 |
|  | 512 | 1e − 5 | 0.795 ± 0.001 | 0.694 ± 0.001 | 0.872 ± 0.003 | 0.314 ± 0.005 | 0.625 ± 0.005 | 0.118 ± 0.133 | 0.124 ± 0.076 | 0.096 ± 0.073 | 0.388 ± 0.121 | 0.079 ± 0.058 | 0.081 ± 0.057 | 0.129 ± 0.148 | 0.114 ± 0.122 | 0.048 ± 0.042 | 0.392 ± 0.012 |
|  |  | 1e − 4 | 0.799 ± 0.001 | 0.709 ± 0.000 | 0.880 ± 0.001 | 0.289 ± 0.006 | 0.638 ± 0.003 | 0.163 ± 0.057 | 0.107 ± 0.044 | 0.159 ± 0.068 | 0.298 ± 0.123 | 0.084 ± 0.034 | 0.157 ± 0.154 | 0.090 ± 0.025 | 0.145 ± 0.021 | 0.094 ± 0.091 | 0.481 ± 0.028 |
|  |  | 5e − 4 | 0.749 ± 0.009 | 0.700 ± 0.000 | 0.857 ± 0.002 | 0.244 ± 0.017 | 0.623 ± 0.003 | 0.074 ± 0.079 | 0.061 ± 0.038 | 0.068 ± 0.038 | 0.426 ± 0.046 | 0.049 ± 0.020 | 0.093 ± 0.063 | 0.187 ± 0.232 | 0.261 ± 0.310 | 0.023 ± 0.014 | 0.218 ± 0.126 |
|  | 1024 | 1e − 5 | 0.778 ± 0.003 | 0.689 ± 0.003 | 0.862 ± 0.002 | 0.296 ± 0.013 | 0.616 ± 0.002 | 0.066 ± 0.046 | 0.051 ± 0.036 | 0.050 ± 0.029 | 0.336 ± 0.145 | 0.032 ± 0.023 | 0.044 ± 0.021 | 0.134 ± 0.136 | 0.154 ± 0.180 | 0.041 ± 0.027 | 0.392 ± 0.008 |
|  |  | 1e − 4 | 0.812 ± 0.002 | 0.713 ± 0.003 | 0.886 ± 0.000 | 0.331 ± 0.012 | 0.648 ± 0.007 | 0.108 ± 0.073 | 0.188 ± 0.146 | 0.237 ± 0.192 | 0.416 ± 0.063 | 0.178 ± 0.127 | 0.185 ± 0.098 | 0.052 ± 0.038 | 0.100 ± 0.092 | 0.088 ± 0.004 | 0.269 ± 0.147 |
|  |  | 5e − 4 | 0.742 ± 0.007 | 0.705 ± 0.002 | 0.856 ± 0.004 | 0.226 ± 0.008 | 0.639 ± 0.010 | 0.168 ± 0.100 | 0.163 ± 0.116 | 0.180 ± 0.152 | 0.402 ± 0.146 | 0.150 ± 0.120 | 0.250 ± 0.076 | 0.373 ± 0.114 | 0.379 ± 0.132 | 0.029 ± 0.009 | 0.421 ± 0.025 |
| 10.0 | 256 | 1e − 5 | 0.690 ± 0.024 | 0.698 ± 0.002 | 0.828 ± 0.015 | 0.197 ± 0.020 | 0.614 ± 0.013 | 0.118 ± 0.079 | 0.161 ± 0.140 | 0.158 ± 0.180 | 0.333 ± 0.166 | 0.147 ± 0.147 | 0.132 ± 0.091 | 0.324 ± 0.129 | 0.328 ± 0.148 | 0.040 ± 0.021 | 0.485 ± 0.068 |
|  |  | 1e − 4 | 0.492 ± 0.017 | 0.704 ± 0.007 | 0.736 ± 0.002 | 0.085 ± 0.015 | 0.596 ± 0.005 | 0.201 ± 0.061 | 0.349 ± 0.103 | 0.351 ± 0.085 | 0.350 ± 0.148 | 0.275 ± 0.085 | 0.185 ± 0.147 | 0.279 ± 0.208 | 0.279 ± 0.214 | 0.029 ± 0.020 | 0.351 ± 0.056 |
|  |  | 5e − 4 | 0.462 ± 0.003 | 0.699 ± 0.007 | 0.713 ± 0.003 | 0.048 ± 0.007 | 0.611 ± 0.014 | 0.225 ± 0.149 | 0.356 ± 0.249 | 0.327 ± 0.228 | 0.170 ± 0.090 | 0.268 ± 0.183 | 0.274 ± 0.035 | 0.417 ± 0.054 | 0.402 ± 0.060 | 0.027 ± 0.004 | 0.474 ± 0.182 |
|  | 512 | 1e − 5 | 0.728 ± 0.027 | 0.698 ± 0.005 | 0.841 ± 0.012 | 0.227 ± 0.040 | 0.610 ± 0.013 | 0.114 ± 0.096 | 0.198 ± 0.084 | 0.210 ± 0.092 | 0.340 ± 0.155 | 0.157 ± 0.068 | 0.107 ± 0.052 | 0.225 ± 0.158 | 0.209 ± 0.129 | 0.038 ± 0.025 | 0.411 ± 0.041 |
|  |  | 1e − 4 | 0.513 ± 0.013 | 0.700 ± 0.006 | 0.743 ± 0.006 | 0.109 ± 0.034 | 0.604 ± 0.034 | 0.144 ± 0.113 | 0.269 ± 0.202 | 0.281 ± 0.209 | 0.315 ± 0.135 | 0.213 ± 0.159 | 0.272 ± 0.156 | 0.288 ± 0.157 | 0.250 ± 0.131 | 0.050 ± 0.015 | 0.440 ± 0.091 |
|  |  | 5e − 4 | 0.497 ± 0.014 | 0.701 ± 0.003 | 0.723 ± 0.001 | 0.091 ± 0.002 | 0.636 ± 0.016 | 0.191 ± 0.176 | 0.287 ± 0.251 | 0.275 ± 0.224 | 0.204 ± 0.106 | 0.210 ± 0.170 | 0.122 ± 0.096 | 0.167 ± 0.168 | 0.148 ± 0.172 | 0.026 ± 0.011 | 0.449 ± 0.089 |
|  | 1024 | 1e − 5 | 0.699 ± 0.023 | 0.693 ± 0.003 | 0.821 ± 0.011 | 0.198 ± 0.024 | 0.600 ± 0.010 | 0.179 ± 0.133 | 0.242 ± 0.117 | 0.295 ± 0.155 | 0.362 ± 0.161 | 0.232 ± 0.118 | 0.119 ± 0.082 | 0.181 ± 0.151 | 0.127 ± 0.087 | 0.027 ± 0.013 | 0.389 ± 0.025 |
|  |  | 1e − 4 | 0.499 ± 0.012 | 0.704 ± 0.001 | 0.735 ± 0.005 | 0.067 ± 0.014 | 0.595 ± 0.011 | 0.168 ± 0.018 | 0.243 ± 0.056 | 0.239 ± 0.075 | 0.343 ± 0.150 | 0.187 ± 0.047 | 0.217 ± 0.114 | 0.387 ± 0.142 | 0.403 ± 0.132 | 0.038 ± 0.020 | 0.510 ± 0.091 |
|  |  | 5e − 4 | 0.500 ± 0.011 | 0.706 ± 0.009 | 0.731 ± 0.003 | 0.108 ± 0.013 | 0.617 ± 0.017 | 0.177 ± 0.089 | 0.276 ± 0.136 | 0.263 ± 0.115 | 0.306 ± 0.152 | 0.214 ± 0.113 | 0.220 ± 0.149 | 0.328 ± 0.229 | 0.308 ± 0.215 | 0.025 ± 0.008 | 0.449 ± 0.039 |

Table 13: CITRIS hyperparameter search (3 random seeds). Empty cells indicate failed runs due to training diverging.

| lat. dim. | batch size | lr | $R^2$ diag ↑ | $R^2$ sep ↓ | Spearman diag ↑ | Spearman sep ↓ |
|---|---|---|---|---|---|---|
| 16 | 128 | $1e-4$ | $0.106 \pm 0.028$ | $0.241 \pm 0.020$ | $0.222 \pm 0.025$ | $0.406 \pm 0.032$ |
| | | $1e-3$ | $0.072 \pm 0.076$ | $0.308 \pm 0.095$ | $0.225 \pm 0.110$ | $0.464 \pm 0.092$ |
| | | $1e-2$ | — | — | — | — |
| | 512 | $1e-4$ | $0.193 \pm 0.024$ | $0.207 \pm 0.064$ | $0.360 \pm 0.024$ | $0.398 \pm 0.085$ |
| | | $1e-3$ | $0.117 \pm 0.058$ | $0.276 \pm 0.043$ | $0.271 \pm 0.063$ | $0.440 \pm 0.042$ |
| | | $1e-2$ | — | — | — | — |
| | 2048 | $1e-4$ | $0.157 \pm 0.038$ | $0.236 \pm 0.025$ | $0.370 \pm 0.072$ | $0.464 \pm 0.043$ |
| | | $1e-3$ | $0.142 \pm 0.076$ | $0.236 \pm 0.028$ | $0.301 \pm 0.082$ | $0.410 \pm 0.056$ |
| | | $1e-2$ | $0.112 \pm 0.059$ | $0.251 \pm 0.026$ | $0.260 \pm 0.086$ | $0.407 \pm 0.018$ |
| 32 | 128 | $1e-4$ | $0.051 \pm 0.073$ | $0.204 \pm 0.042$ | $0.255 \pm 0.080$ | $0.425 \pm 0.038$ |
| | | $1e-3$ | $0.084 \pm 0.110$ | $0.325 \pm 0.081$ | $0.249 \pm 0.087$ | $0.496 \pm 0.089$ |
| | | $1e-2$ | — | — | — | — |
| | 512 | $1e-4$ | $0.065 \pm 0.054$ | $0.179 \pm 0.009$ | $0.222 \pm 0.047$ | $0.385 \pm 0.013$ |
| | | $1e-3$ | $0.002 \pm 0.023$ | $0.307 \pm 0.006$ | $0.189 \pm 0.019$ | $0.482 \pm 0.012$ |
| | | $1e-2$ | — | — | — | — |
| | 2048 | $1e-4$ | $0.101 \pm 0.047$ | $0.221 \pm 0.059$ | $0.306 \pm 0.038$ | $0.438 \pm 0.057$ |
| | | $1e-3$ | $0.055 \pm 0.021$ | $0.242 \pm 0.036$ | $0.228 \pm 0.022$ | $0.441 \pm 0.069$ |
| | | $1e-2$ | $-0.016 \pm 0.007$ | $0.215 \pm 0.005$ | $0.168 \pm 0.006$ | $0.404 \pm 0.026$ |

