# OpenReview forum: "Sanity Checking Causal Representation Learning on a Simple Real-World System"
_ICML.cc/2025/Conference — ICML 2025 oral_

### Official Review · Reviewer_uuRR · 2025-03-11

**Overall Recommendation:** 1

**Summary:**

This work evaluates causal representation learning (CRL) methods on a simple, real-world optical system designed as a sanity check for CRL assumptions. The authors argue that while many CRL methods show theoretical promise, they fail when applied to this controlled real-world system due to critical issues such as noise sensitivity and unrealistic assumptions about the mixing function. To investigate further, they perform a synthetic ablation using a deterministic simulator for the optical experiment, revealing that many methods also fail on simplified synthetic data. They evaluate representative methods from contrastive, multiview, and time-series CRL approaches, highlighting reproducibility challenges and performance gaps. Extensive experiments on this benchmark reveal that existing CRL techniques struggle to recover the underlying causal factors, underscoring the gap between theory and real-world applicability.

## update after rebuttal

After the discussion with the authors, it seems the key message and motivation of this work are not clear, making the implications of the results in this work limited. Therefore, I am leaning not to accept this work.

**Claims And Evidence:**

Yes

**Essential References Not Discussed:**

N/A

**Experimental Designs Or Analyses:**

Yes

**Methods And Evaluation Criteria:**

MCC may not be a stable evaluation metric for non-linear correlations. It would be nice to extend to more evaluation metrics to robusitify the evaluation results.

**Other Comments Or Suggestions:**

N/A

**Other Strengths And Weaknesses:**

## Strengths

(+) This paper considers a critical point in CRL, which has been neglected by the advance of theories in this field.

(+) The authors conduct extensive and thorough discussion about their experimental settings, and provide interesting results;

## Weakness

(-) The study lacks systematicity:
- From the design of testbed, it considers only a limited part of the factors in CRL such as the mixing function:
  - In theory, especially, there are already some works working on measurement errors such as [1]. The focus of CCRL seems not to resolve the measurement error introduced in the mixing functions.
  - Moreover, there are also other factors such as hidden confounders, missing data. What factors do the authors choose to study when constructing the benchmark and why? Are they sufficiently representative?
  - In implementation, it has already been mentioned that network architectures and optimization could also affect CRL. However, they are not sufficiently discussed and studied.
- From the methods: there are also other CRL works open sourcing codes. Why do the authors choose the current three methods instead of the others?
- In addition, it is worthwhile to clearly highlight the contributions of the benchmark from the existing CRL benchmarks.

(-) The motivation and the key messages are not clear:
- In the analysis of CCRL, the noises in mixing function seem to be an critical issue for the failure of CCRL, while it is not the focus of CCRL;
- In the analysis of Multiveiw CRL, noises in mixing function seem not be influential anymore, while the separation of latent variables seem to be a critical influencing factor;
- In the analysis of CITRIS, it is even confusing that the implementation of the submodules may be the reason for its failure.
- It would be better to formulate the underlying research problem and the concerned factors that this benchmark is designed to study and control the other factors more rigorously. Otherwise, it is quite confusing to see what messages this work would like to convey through benchmarking.

(-) The technical novelty may be limited since it is directly applied from Gamella et al. 2025.


**References**

[1] Causal Discovery with Linear Non-Gaussian Models under Measurement Error: Structural Identifiability Results

**Questions For Authors:**

Please find my questions in the previous sections.

**Relation To Broader Scientific Literature:**

N/A

**Theoretical Claims:**

N/A

---

> ### Author Rebuttal · Authors · 2025-03-30
>
> Thank you for your review. We respond to your points below.
>
> **Motivation & differences to other benchmarks:** As all other reviewers highlight and we stress in the paper, the primary distinction of our sanity check is that it stems from a real, physical system to evaluate the practical validity of assumptions:
> - Rev. DQvb: "[evaluating on synth. data] provides further validation for the theoretical foundations of [CRL] methods but yields limited insight into their applicability to real-world problems. This point seems to be lost on many researchers, and the authors of this paper place it front and center. Bravo."
> - Rev. KaPR: "the data is highly desirable and useful for CRL evaluation"
> -Rev. 55Kr : “I believe this is the first benchmarking of CRL algorithms on data generated by a physical process.” and “would be a much preferable benchmark for practical purposes than what is currently in the literature”
>
> **Key messages:** You say “It would be better to formulate the underlying research problem and the concerned factors that this benchmark is designed to study and control the other factors more rigorously.” We believe we have clearly explained the factor we study (the mixing function), we rigorously control the other factors, e.g., generating the latents according to each model’s assumptions, and made very clear what conclusions can be drawn. Citing Rev. DQvb: “The claims are clearly delineated, and the authors take some pains to make clear what their "sanity check" cannot do”
>
> Regarding failure in the optimization routines, we have performed additional experiments to study this mode of failure. See the answer to reviewer Rev. 55Kj for a draft of what will be included in the final version.
>
> **MCC:** As Rev. DQvb points out, we aim to “meet each method on their own grounds”, using the same validation metrics as in the original papers. As we stress, this is a sanity check, and our focus is not to compare methods to each other (difficult given their different goals). We agree that the MCC is not an ideal metric for CRL, but it is the most commonly used one and we are not aware of a metric that directly addresses its shortcomings. It is well beyond the scope of this study to derive a novel metric.
>
> **Systematicity:**
> - Measurement error: The work you reference considers measurement errors for causal discovery, which we do not directly see being applicable to CRL. While CCRL does not explicitly consider measurement noise, this is no reason to exclude it from the sanity check. Measurement noise is inherent to any real-world data; the failure of this method points to an important issue in realistic scenarios that should perhaps be considered by future works.
> - Other failure modes: As you point out, many things can go wrong—beyond the mixing function—when transferring CRL from theory to practice. However, we argue that focusing precisely on this single factor (as opposed to considering everything at once) is what gives our study systematicity (see answer to Rev. 55Kj). As Rev. 55Kj points out, we meet all other assumptions of each method to isolate and better understand the effect of misspecified assumptions regarding the mixing function. Given the subsequent failure of methods after only this single (!) misspecification, *we believe our setting already highlights a problem worth discussing*. Including more factors of analysis is of course always possible, but (1) it is beyond the scope of such a paper and (2) it dilutes the point of focusing (and understanding) the failure mode in such a simple setting.
> - As you point out, this work is a sanity check and *not* a benchmark that aims to exhaustively study any possible mode of failure. We stress this throughout, as other reviewers (DQvb) also acknowledge. CRL is far from being practically applicable, and our sanity check is meant to highlight the “remarkable failure of SOTA systems in such a simple setting” (Rev. DQvb), rather than comparing working approaches.
>
> **Method choice:** Any choice will exclude other methods, we don’t claim exhaustiveness, nor do we draw conclusions from these representatives to their whole family of methods. Our results highlight their failures and that this is worth looking into. Citing Rev. 55Kj: “[it] convinces me that existing CRL algorithms, at least out of the box, are inadequate for practical use”
>
> We chose our representative methods for the following (often pragmatic) reasons:
> - CCRL: allows for general nonlinear mixing functions (cf. Ahuja et al. 2023, Zhang et al. 2023 with polynomial mixing, Squires et al. 2023 with linear mixing) and uses a contrastive loss, which has proven superior to an encoder-decoder in other representation learning domains.
> - Multiview CRL: multiview approach that permits more than two views and considers nonlinear mixing (on images), with accessible code.
> - CITRIS: considers complex nonlinear mixing experiments (images), very well maintained code base.
>
> **Technical novelty:** See our answer to Rev. KaPR.

---

> > ### Comment · Reviewer_uuRR · 2025-04-07
> >
> > Thank you for the detailed explanation and apologies for the delay in my reply. With all due respect to the comments from other reviewers, I also appreciate the benchmarking from real-world process, as it is necessary for the development of CRL. However, I found the key message delivered in the current manuscript remains confusing.
> >
> > I have also checked the authors' response to Reviewer 55Kj. It turns out, all of us agree that there exist two critical reasons for the failures of the CRL, i.e., misspecification and optimization.
> > - For a benchmark with real-world data, both of them matters, while it is straightforward that misspecification will lead to failures. Especially, the mixing function in the real-world data from the benchmark inherently contains noises in the mixing function, or in the measure mement. Hence, `We believe we have clearly explained the factor we study (the mixing function), we rigorously control the other factors, e.g., generating the latents according to each model’s assumptions, and made very clear what conclusions can be drawn.` may not be interesting.
> > - Not to mention that **the other factors are not rigorously controlled** (i.e., optimization).
> > - Although I also appreciate the additional hyperparameter tuning experiments, the failures of the CRL methods selected for the last two categories on both synthetic data and realistic data seem to make the key message that this paper attempts to deliver even more confusing. If we can not identify the exact failure reasons for those methods, how can we learn from the benchmarking results and the failures? If the authors focus on the misspecification issue, then more CRL methods shall be benchmarked and demonstrate the success on the synthetic data.
> > - From the solution side, there seem already some existing works dealing with the misspecification issue, e.g., the referred one in my original review, and many. If the focus is the misspecification issue, it seems not interesting.

---

> > > ### Author Response · Authors · 2025-04-09
> > >
> > > Thank you for your additional comments.
> > >
> > > **Re: “not interesting”**. We respectfully disagree, and we believe that the other reviewers, given their reviews, do as well. Our study is the first that presents a sanity check for CRL using a real-world system, where the ground truth is known. After reviewing our results, it may no longer be particularly surprising that most methods fail in such a setting, but we are the first to actually apply methods to a real-world dataset and collect evidence of this failure. The work you cite regarding misspecification deals with causal discovery, and it is not at all obvious how insights from misspecified models in that domain translate to causal representation learning (especially the mixing function).
> > >
> > > **Re: “other factors not controlled”**. We again disagree. As we have explained above, we have designed our experiments with extreme care,  to control for all modelling factors except the mixing transformation (our experimental target). We have gone to significant lengths to control for many points of failure w.r.t. optimization (pipeline sanity checks, direct contact with original authors, hyperparameter searches, etc.).

---

### Official Review · Reviewer_55Kj · 2025-03-12

**Overall Recommendation:** 4

**Summary:**

The authors test three representative causal representation learning (CRL) algorithms on real-world data generated from Causal Chambers, which is a small light tunnel that takes in 5 controllable inputs (factors) and outputs numerical sensor and imaging data. The authors treat these measurement processes as a noisy mixing function of the factors and evaluate whether CRL algorithms are capable of recovering them up to   indeterminacies as promised by their respective claims of identifiability. For each representative CRL algorithm, the factors are sampled accordingly to satisfy the respective latent assumptions. As an ablation, synthetic versions of the measurements are obtained by simulating the known physical laws (for the sensors) or by mimicking supervised examples with an MLP (for the images). The results show that, despite latent assumptions being perfectly satisfied, none of the CRL algorithms are able to recover the true factors up to the desired indeterminacies from this noisy real-world mixing function. The same conclusion holds for the synthetic ablation, with the exception of the contrastive CRL method.

**Claims And Evidence:**

The submission introduces a new benchmark for CRL. It claims to be the first sanity check for CRL based on data from a physical process, to my knowledge this is true. I agree with the authors that this would be a much preferable benchmark for practical purposes than what is currently in the literature. The submission convinces me that existing CRL algorithms, at least out of the box, are inadequate for practical use. However, I'm a bit unsure about whether the experiments convincingly capture the failure modes: see the next section for details.

**Essential References Not Discussed:**

I believe this is the first benchmarking of CRL algorithms on data generated by a physical process. It includes all the necessary references to understand the contribution.

**Experimental Designs Or Analyses:**

The data generation part of the design is great, I really enjoyed how the authors mimicked the latent processes as per the original papers before passing it to the physical simulator. The synthetic ablation is a nice touch. As mentioned above I think they could have done more to investigate the optimization landscape.

I also think taking only one representative from the classification of CRL methods may not be sufficient. Currently they appear to be based on the constraints under which they obtain identifiability (which, fair enough, this determines how the data should be generated), but they can actually be algorithmically speaking quite different. As an example for interventional CRL, Zhang et al., 2023 or Ahuja et al., 2023 use an autoencoder setup instead of the contrastive approach. It would be interesting to see how these differ in practice (Ahuja et al. 2023 also consider do-interventions which are qualitatively different but easy to implement in this setting).

**Methods And Evaluation Criteria:**

I think the authors are missing a crucial discussion in their evaluation: optimization.

The authors state that the set-up is "geared towards evaluating the assumptions concerning the mixing function" (l. 97r). I think this amounts to saying that the generative process assumed by CRL methods are mis-specified for the true physical process here. While I don't doubt that this could be true, the evaluation here does not rule out bad local optima, which I think is the other possible failure mode here. For example the authors note that results differ drastically over the five random initializations in all methods.

I find it confusing that the authors emphasized using the original implementations, as hyperparameters used in their synthetic experiments for example should certainly not be expected to perform well in a new setting without any tuning. I think the experiments would be much more convincing if more effort were made to ensure that the algorithm at least can reach some sort of stable optima. Of course, if this is simply not possible given the loss landscape, then that itself is a failure mode as well and should be discussed (maybe we should advocate for simpler models then for CRL).

**Other Comments Or Suggestions:**

### Suggestions

- For general audiences, I think the paper could do a better job explaining what the theoretical guarantees, and thus resulting failure modes, of the CRL algorithms are. Usually, it is based on minimizing some objective relative to the true generative process which is assumed to have certain properties, and based on minimizing this objective exactly, saying that the resulting representations are related to the ground truth in some sense. Given the theory, the only failure modes that I can think of when transitioning to practice is to do with 1) mis-specification of the generative process, and 2) optimization issues. As mentioned above, I think the paper explains 1), but not 2).

**Other Strengths And Weaknesses:**

### Strengths

- The paper is wonderful to read as someone already familiar with the theoretical literature.
- The study is extremely important at this stage of the literature for CRL. Continued experimentation and progress based on a benchmark like this can possibly determine whether CRL becomes a practical tool or remains a theoretical exercise in the future.

### Weaknesses

- There may be some novelty overlap with the original causal chambers paper (Gamella et al., 2025), which seems to include an ICA experiment. If the focus here is on strictly causal settings maybe it would be good to mention the difference.

**Questions For Authors:**

I don't have any additional questions, but I will just summarize the two points I tried to make which I think would improve the paper. As-is my score is "Weak Accept" which is primarily based on clarity and the importance of the problem.

1. Discussion of failure modes of CRL relative to the theory, in particular discussing failed optimization in addition to unmet assumptions. Empirically, attempting to improve optimization in the experiments or showing that it is infeasible for a given algorithm (see "Methods And Evaluation Criteria" and "Suggestions").

2. Choosing representatives for CRL algorithms to evaluate not only based on their assumptions for identifiability but also practical implementation (e.g., autoencoder vs contrastive, see "Experimental Designs Or Analyses").

I would be happy to argue for acceptance if the authors could satisfactorily discuss aspects of (1). If additional experiments could eventually be included in the paper corresponding to (1) and (2) I would lean towards strong acceptance.


POST REBUTTAL COMMENT:

Thank you for responding to my questions. As mentioned, the discussion about optimization is what moves the needle most for me. The author's response on architecture, sample sizes, etc. should be included/expanded upon in the paper. Therefore I will raise my score to 4. I understand it is unfeasible to ensure a high quality for new implementations within the rebuttal period but I still feel the experiments could have been more comprehensive from the start, thus I cannot justify a 5.

**Relation To Broader Scientific Literature:**

CRL is crucial for causal reasoning over unstructured data, which is of great importance in many scientific areas, e.g., biology, physics, medicine. To date most advancements in this area have been theoretical (in particular focused on identifiability), so it is very important to design benchmarks that support the design of CRL algorithms for practical uses.

**Theoretical Claims:**

N/A

---

> ### Author Rebuttal · Authors · 2025-03-30
>
> Thank you for your careful review and constructive feedback.
>
> **Re: Gamella et al., 2025:** Thank you for raising this point. As other reviewers (KaPR, uuRR) also pointed this out, we will add a paragraph text in section 2 to clearly separate our contributions from Gamella et al., 2025. Please see our response to reviewer KaPR below for the details of our contribution w.r.t. those of Gamella et al., 2025.
>
> **Re: additional methods:** We agree in principle that running our sanity check on more methods would be insightful. However, our approach of meeting methods in their own ground (as praised by Reviewer DQvb)—using the original code and evaluation techniques—meant that getting a method to work takes a significant amount of time and effort. We can use the extra page for this, but given how difficult it was to get the current methods to run (incl. communicating with the authors + tests to ensure no bugs) we don’t want to promise that we can do this for Zhang et al., 2023 or Ahuja et al., 2023. It is definitely not feasible before the rebuttals are due. We hope others will use our work to investigate individual methods in full detail in future works.
>
> **Re: optimization:** Thank you for raising this point; we find your categorization of failure modes (misspecification vs. optimization) very useful. We fully agree that the latter is often overlooked and that an additional discussion will improve the paper. To support it:
>
> 1. We have now performed additional experiments doing a large hyperparameter search for each method on the real data. The results ([see here](https://blue-gina-78.tiiny.site/)) show that the hyperparameter choice does not significantly affect the final metrics, across all models. We will repeat the experiments on the synthetic data for the camera-ready version as time constraints did not permit doing so for the rebuttal.
>
> 2. We will add the training curves of all methods ([see here](https://maroon-agneta-85.tiiny.site/)) to the appendix. From them, we conclude that training does converge (although noisy for some methods, cf. Multiview CRL), not raising immediate concerns about optimization failing in some catastrophic way.
>
> With the above we hope to address (and exclude) some of the key potential failure modes regarding optimization, i.e., losses not going down during training or hyperparameter choices greatly affecting the outcome. Our hyperparameter search was possible because we have access to ground truth labels to compute the final metrics and perform tuning. However, in a practical setting without ground truth, model selection of unsupervised models is still a great challenge, and finding the modes of failure w.r.t. optimization would be even more difficult.
>
> In addition, we put in a substantial effort to exclude bugs and other issues by performing preliminary checks to reproduce the synthetic experiments in the original  papers using our pipeline, as well as consulting the authors of the methods to advise us, where possible.
>
> Regarding a misspecification in the generative process: to isolate the mixing transformation as the only source of mismatched assumptions, we generated the data closely following the assumptions for each method. We do not merely follow the general assumptions, we also closely replicated the more subtle details of the respective methods’ data generating processes, such as no. of latent variables, noise mean and variances, intervention strengths etc.
>
> To summarize:
> - CCRL appears to fail because of a misspecification of the generative process (noiseless vs. noisy mixing function), since the method works well for the synthetic ablation but fails on the real data.
>
> - Because the other methods fail on both real and synthetic data, it seems like failure is instead also due to optimization issues. Because the training curves converge, and the performance does not change given different hyperparameters, we argue that there are either architecture choices that need deeper investigation, or finite sample issues that should be better understood. All identifiability results in CRL rely on the unrealistic assumption of infinite data, but how much data is enough in practice remains elusive. We collected as many samples for each method as was possible given the scope of this study, but because there are actual costs to collecting real data samples, these still fall short of the enormous sample sizes in the original papers: CCRL 10k vs 25k (per env.), Multiview 60k vs 145k, CITRIS 100k vs 100-250k.
>
> We will add all the above as additional discussion using the extra page in the final version. This will be framed (as you suggest) by an explanation for general audiences of how the theoretical guarantees of CRL methods work and how the two possible modes of failure result from this.

---

### Official Review · Reviewer_KaPR · 2025-03-13

**Overall Recommendation:** 5

**Summary:**

The paper benchmarks 3 representative causal representation learning methods on real data produced by a simple controlled physical system with known ground truth. CRL models have underlying causal factors that are "mixed" into observed variables. In this benchmark, the underlying causal model is simulated (Line 145), but the mixing is done by the physical system. In addition to releasing public benchmarking data and evaluating key CRL methods, the paper highlights some methodological and reproducibility issues in CRL research.

**Claims And Evidence:**

Yes.

**Essential References Not Discussed:**

I would find it helpful if the paper more clearly delineated its contributions---and especially the physical system configuration and resulting datasets---compared to the Gamella et al. (2025).

**Experimental Designs Or Analyses:**

Yes, I checked the experimental design, data generation process, and the analyses. I'm not familiar with such physical systems, but the process seems quite reasonable and the data is highly desirable and useful for CRL evaluation. It's too bad the causal model has to be simulated, but anyway the mixing (which is done by the physical system) is what's crucial for CRL and the current benchmark data is nevertheless challenging and interesting enough.

**Methods And Evaluation Criteria:**

Yes, a reasonable balance is struct between method-specific evaluations and method-agnostic comparisons.

**Other Comments Or Suggestions:**

- fix "bayesian" (Line 506R and 557L) in references and check for any other similar mistakes

**Other Strengths And Weaknesses:**

- extremely polished and well written
- expected to be a common benchmark for much of the future work in CRL

**Questions For Authors:**

N/a

**Relation To Broader Scientific Literature:**

Details of the physical system and other real datasets from it are recently published in Nature (Gamella et al. 2025). This complements that with CRL-specific datasets, benchmarking, and methodological discussion.

Also a nice complement to the the Sachs et al. datasets, which have been widely used in causal discovery benchmarking but are not suitable for CRL.

**Theoretical Claims:**

N/a

---

> ### Author Rebuttal · Authors · 2025-03-30
>
> Thank you very much for your positive review! We answer your points below.
>
> **Re: bayesian:** We have fixed this in an updated version of the manuscript, thank you.
>
> **Re: Contributions w.r.t.. Gamella et al., 2025:** Indeed, Gamella et al., 2025 introduce the light tunnel. In our work we leverage it to construct a meaningful benchmark for CRL. This required an in-depth analysis of the experimental setup in light of CRL assumptions (section 2), designing new experiments and collecting the data, and developing the deterministic simulators for the synthetic ablation. None of these existed in Gamella et al., 2025. The ICA experiment in Gamella et al., 2025 could also be used as a first test for CRL, but since the latent factors are all independent this didn’t seem very interesting. We will add a paragraph in section 2 to clearly separate our contributions.

---

### Official Review · Reviewer_DQvb · 2025-03-16

**Overall Recommendation:** 4

**Summary:**

The authors evaluate existing methods for causal representation learning (CRL) on a real-world system that appears to meet the assumptions of such methods, and yet they show that the methods almost entirely fail to recover a valid causal model of the system. Specifically, they test methods under assumptions about the mixing function that transforms the inputs (“causal factors”) into observations, focusing on cases in which such assumptions underpin the identifiability results of the methods.

## Update after Rebuttal

I remain convinced that this paper should be accepted. It is an impressive and useful contribution to the very difficult problem of empirical evaluation of methods for causal representation learning.

**Claims And Evidence:**

One of the primary benefits of empirical evaluation is that it can evaluate whether the assumptions of a given method are realistic enough to provide useful results. As the authors clearly state: “This practice [evaluating on synthetic data] provides further validation for the theoretical foundations of these methods but yields limited insight into their applicability to real-world problems.” This point seems to be lost on many researchers, and the authors of this paper place it front and center. Bravo.

**Essential References Not Discussed:**

I don’t know of references missing, though I am not an expert in this area. The authors cite a large number of apparently relevant work.

**Experimental Designs Or Analyses:**

The experiments apply each of the three methods to data drawn from the physical system (a light tunnel) that forms the centerpiece of this work. As noted above, the diversity of the methods evaluated virtually necessitates a diverse set of experiments, and the authors fairly clearly describe the different approaches used.

One of the few disappointing aspects of this paper is that it offers only hypotheses, rather than strong evidence, for why the differences in data (between synthetic and real) produce differences in performance. The most promising case is contrastive CRL (CCRL), in which the substitution of highly similar, but synthetic, data produces far better results than for data drawn from the actual system. Even in this case, however, the authors end on a hypothetical note, saying: "A possible explanation is that CCRL relies on detecting interventions in the latent space, a sensitive statistical problem for which it may lack power in the case of a noisy mixing process.” In the two other cases, however, the authors offer even less clear explanations. For example, in the case of CRL from temporal intervened sequences, the authors state that “…the exact reason for the failure in this setting remains elusive.”

While I don’t think strong experimental evidence about the cause of the observed performance difference is mandatory, it would provide greater confidence that the failure of each CRL method is based on a limitation of the method itself rather than some error in the way the authors configured or evaluated the method. Despite this, the paper is sufficiently interesting, detailed, methodologically solid, and well-written that it deserves publication.

**Methods And Evaluation Criteria:**

Given the diversity of the three methods being evaluated, the methods and evaluation criteria are similarly diverse. The authors are clear about this, but it leads to some difficulty for readers who are not familiar with each of the methods. That said, the authors deserve credit for “meeting the methods on their own ground” rather than attempting to shoe-horn all the methods into a common evaluation framework.

**Other Comments Or Suggestions:**

(None)

**Other Strengths And Weaknesses:**

The authors cleverly call their real-world system a “sanity check” rather than a “benchmark” or other name that implies a domain of realistic complexity. Instead, this domain is remarkably simple (as the authors emphasize), and thus the failure of current state-of-the-art methods is even more striking.

The claims are clearly delineated, and the authors take some pains to make clear what their "sanity check" *cannot* do (second-to-last paragraph of Section 1). They also clearly outline the different sources of noise. The authors also take substantial pains to test for bugs in their analysis pipeline (see the last paragraph before section 3.1).

**Questions For Authors:**

(None)

**Relation To Broader Scientific Literature:**

The paper lies at the frontier of work on causal representation learning, a vitally important research area, given the problems with current, largely associational models that learn representations. The key issue that this paper explores is whether the identifiability assumptions of current methods are practical. While this is only one case, it still represents an important step forward in evaluating methods for CRL.

**Theoretical Claims:**

The theoretical claims are those made by the original papers, rather than the ones made by this paper.

---

> ### Author Rebuttal · Authors · 2025-03-30
>
> Thank you for your positive review!
>
> We wanted to be cautious about making unsubstantiated claims regarding the failure points of the algorithms. Due to their complexity (especially Multiview CRL and CITRIS), understanding the exact source of failure may be very difficult, and it may not be possible to pin it down to a single assumption or implementation choice.
>
> However, following your feedback and that of reviewer 55Kj, we have performed additional experiments (hyperparameter searches) for all considered methods and, if accepted, we will add further discussion using the extra page. You can see a draft of this discussion and the supporting experiments in our answer to reviewer 55Kj below (we don’t repeat it here for lack of space).
>
> Thank you for raising the above point; we agree that such a discussion can improve the quality of the paper.

---

### Decision · Program_Chairs · 2025-05-01

**Decision:**

Accept (oral)

**Comment:**

The main purpose of this submission is to benchmark 3 CRL methods using a real physical system from Gamella et al (2025). This is an important problem given the primarily theoretical leaning of CRL papers. As with any real-world experiments, there are necessarily some limitations: Several reviewers pointed out a blind spot with respect to how the authors control the optimizer, as well as significant overlap with Gamella et al (2025). The authors have promised to address both these points, which should be easy. Another reviewer pointed out a misspecification issue, which the authors did not respond to but should acknowledge and discuss in the camera ready.

Although the experiments have some limitations, this is easily the most thorough and realistic sanity check of CRL available to date. Given the timeliness of this submission and overall positive reception from reviewers, I recommend accept.